# Using Noise to Help Reach Global Minima: Turning Matrix Completion into Noisy Matrix Sensing

## Abstract

Matrix completion (MC) is an important yet challenging non-convex problem. In realistic settings, exact recovery of $M^*$ typically requires strong incoherence and an impractically large number of observed entries. Instead of enforcing exact recovery, we inject noise perturbation to construct a closely related surrogate that turns MC into a noisy matrix sensing problem with a more benign landscape. Although this surrogate permits a slight, controllable loss in accuracy, it can be solved effectively via over-parameterization (increasing model size), echoing modern machine learning practices where large models and stochasticity (e.g., SGD, dropout) make hard objectives tractable. Under the assumption that each entry of the matrix is observed independently and uniformly, we establish explicit accuracy–probability trade-offs as functions of the sampling rate $p$ and a user-chosen noise level. Empirically, our approach succeeds in low-observation regimes where classical exact-recovery pipelines are brittle. More broadly, our approach underscores a general paradigm in which noise perturbations are combined with large models to tackle modern ML tasks, and we use MC as a clean benchmark to formalize this perspective and unify noise, over-parameterization, and recoverability within a single framework.

## 1 Introduction

Non-convex optimization presents significant challenges in modern machine learning, particularly in training complex models like deep neural networks, generative models, and beyond. Unlike convex problems, non-convex optimization landscapes are characterized by multiple local minima, saddle points, and regions of flat curvature, complicating the search for global optima (Jain and Kar, 2017). This complexity often leads to convergence issues, where algorithms may become trapped in suboptimal solutions, hindering model performance and generalization capabilities. While practitioners today typically employ stochastic gradient descent (SGD) based algorithms such as ADAM (Kingma and Ba, 2014) with the hope of randomly escaping poor critical points, there remains a critical need for more structured approaches that leverage landscape characterization to significantly enhance the performance of non-convex optimization methods (Sun and Luo, 2019). Consequently, a more in-depth and comprehensive study of such landscapes is essential. One popular way of doing so is to focus on mathematically structured non-convex problems like low-rank matrix recovery. Such problems have garnered increased attention due to their potential to provide deep insights. Low-rank matrix tasks, including matrix completion (MC) and matrix sensing (MS), are crucial in numerous domains like machine learning and signal processing. They involve reconstructing a low-rank matrix from incomplete observations or linear measurements, with applications spanning collaborative filtering in recommendation systems (Koren et al., 2009), motion detection (Fattahi and Sojoudi, 2020), and power system state estimation (Zhang et al., 2017; Jin et al., 2019), to image recovery (Gu et al., 2014) and biomedical imaging (Lustig et al., 2008). More significantly, as these frameworks can encapsulate any polynomial optimization problem (Molybog et al., 2020) and are equivalent to training two-layer quadratic neural networks (Li et al., 2018), their theoretical impact extends well beyond their direct applications in the broader machine learning community.

Matrix sensing generally involves recovering a matrix from a set of linear measurements, formulated as:

$$\min_{X \in \mathbb{R}^{n \times r_{\text{search}}}} \frac{1}{2} \|\mathcal{A}(XX^T) - b\|_2^2 \coloneqq f(XX^T) \quad \text{(MS)} \tag{1}$$

For the sake of convenience, we also denote $h(X) \coloneqq f(XX^\top)$. Here, $\mathcal{A}$ acts on the low-rank matrix $XX^T$ (rank bounded by $r_{\text{search}}$ by construct) and compares it to a vector of observations $b = \mathcal{A}(M^*)$, with $M^*$ being the rank-$r$ ground truth matrix of interest. $\mathcal{A}(\cdot) : \mathbb{R}^{n \times n} \mapsto \mathbb{R}^m$ is a linear function defined as $\mathcal{A}(M) = [\langle A_1, M \rangle, \ldots, \langle A_m, M \rangle]^T$ where $\langle A_i, M \rangle \coloneqq \text{tr}(A_i^\top M)$ and the sensing matrices $\{A_i\}_{i=1}^m$ are given. For simplicity, we assume $r_{\text{search}} = r$. The matrix completion challenge, a special case of matrix sensing, is given by:

$$\min_{X \in \mathbb{R}^{n \times r}} \frac{1}{2} \|\mathcal{A}_\Omega(XX^T - M^*)\|_2^2 \quad \text{(MC)} \tag{2}$$

where $\Omega \subseteq [n] \times [n]$ represents the observed entries of an $n \times n$ matrix. We use the notation $N_\Omega$ to denote the matrix where

$$(N_\Omega)_{i,j} = N_{i,j} \cdot \mathbf{1}_{(i,j) \in \Omega} \tag{3}$$

for any arbitrary $N \in \mathbb{R}^{n \times n}$. $\mathcal{A}_\Omega(\cdot)$ is used to specifically denote the sensing operator of the matrix completion problem, where $\mathcal{A}_\Omega(M) = \text{vec}(M_\Omega)$. Matrix completion is distinguished by its reliance on the sample rate and the matrix's incoherence parameters. These parameters dictate the spread of matrix information across its entries and singular vectors (Candès and Tao, 2010). This dependency complicates matrix completion compared to matrix sensing, where challenges are often more tractable due to properties like Restricted Isometry Property (RIP) (see definition C.1). Below, we introduce RSS and RSC (equivalent to RIP) instead for more notational flexibility:

**Definition 1.1** (Restricted Strong Smoothness (RSS) and Restricted Strong Convexity (RSC))**.** Function $f$ defined in (1) satisfies the $(L_s, r)$-RSS property and the $(\alpha_s, r)$-RSC property if

$$f(M) - f(N) \leq \langle M - N, \nabla f(N) \rangle + \frac{L_s}{2} \|M - N\|_F^2$$

$$f(M) - f(N) \geq \langle M - N, \nabla f(N) \rangle + \frac{\alpha_s}{2} \|M - N\|_F^2$$

are satisfied, respectively for all $M, N \in \mathbb{R}^n$ with $\text{rank}(M), \text{rank}(N) \leq r$. Note that RSS and RSC provide a more expressible way to represent the RIP property Recht et al. (2010), with $\delta_r = (L_s - \alpha_s)/(L_s + \alpha_s)$.

Matrix completion problems, on the other hand, all have missing entries, which means that they could not have a valid RSC constant since the null-space of $\mathcal{A}_\Omega(\cdot)$ is always non-trivial. This is why incoherence condition was proposed to use as a metric to guarantee recovery of completion problems:

**Definition 1.2** ($\mu_0$-incoherence)**.** (Ge et al., 2017) Given a rank-$r$ matrix $M \in \mathbb{R}^{n_1 \times n_2}$, we say it is $\mu_0$-incoherent if its truncated SVD decomposition $U\Sigma V^\top$ satisfies

$$\|e_i^\top U\|_2 \leq \sqrt{\mu_0 r / n_1}, \quad \|e_j^\top V\|_2 \leq \sqrt{\mu_0 r / n_2}$$

$\forall i, j \in [n_1], [n_2]$, where $e_i$ is the $i$-th standard basis of $\mathbb{R}^{n_1}$ and $e_j$ is the $j$-th standard basis of $\mathbb{R}^{n_2}$.

Since $\mu_0$ is hard to gauge prior to solving this problem, applying matrix completion guarantees can be more challenging than matrix sensing problems with valid RIP (therefore RSC and RSS) constants.

## 1.1 Related Works

### Recovery Guarantees

The foundational work by Candès and Recht (2009) established that exact matrix recovery is possible from few entries, requiring a sample size of $\mu_0 n^{1.2} r \log(n)$ for $n \times n$ matrices of rank $r$ with incoherence parameter $\mu_0$. Enhancements in recovery guarantees and computational efficiencies followed, including spectral-gradient descent algorithms by Keshavan et al. (2010) and deeper insights into incoherence by Candès and Tao (2010). Studies by Recht (2011) and Gross (2011) expanded on these by demonstrating successful recovery without uniform random sampling, while Ding and Chen (2020) refined sampling orders further to $\mu_0 r \log(\mu_0 r) n \log(n)$. Another notable approach

| Approach | Conditions for recovery w.h.p. |
| --- | --- |
| Semi-definite Relaxation (Ding and Chen, 2020) | $p \geq \mathcal{O}\left(\mu_0 r \log(\mu_0 r) n \log(n)/n^2\right)$ |
| BM Factorization (Zilber and Nadler, 2022a) | $p \geq \mathcal{O}\left(\mu_0 r \max\{\log n, \mu_0 r \kappa^2\}/n\right)$ |
| Nuclear Norm Lasso (Chen et al., 2020) | $p \geq \mathcal{O}\left(\mu_0^2 \log(n)^3/n\right)$ |
| Spectral Methods (Chen et al., 2021) | $p \geq \mathcal{O}\left(\kappa^2 \mu_0 r \log(n)/n\right)$ |
| Ours | Arbitrary $p$, trades off with accuracy |

Table 1: Brief summary of state-of-the-art recovery guarantees of different approaches in solving (2). This assumes each entry of $M^*$ is sampled uniformly with probability $p$. $\mu_0$ and $\kappa$ denotes incoherence and condition number for $M^*$ respectively.

for the classic matrix completion problem (2) is the non-convex Burer-Monteiro (BM) factorized formulation (Ge et al., 2016; 2017; Du et al., 2017), generally requiring $p \geq \text{poly}(\kappa, r, \mu_0, \log n)/n$, with recent works like Zilber and Nadler (2022a) offering progressively sharper bounds under this framework. Other formulations include the convex matrix lasso that uses nuclear norm penalization (Chen et al., 2020) and elementary spectral methods (Chen et al., 2021). If we assume that each entry of the desired ground truth matrix $M^*$ is sampled independently with probability $p$, we could then summarize the different requirements for successful recovery with high probability for these techniques in Table 1. Note that we only included state-of-the-art recovery guarantees. From a high-level, existing requirements of $p$ are only described up to an unspecified constant, so the true $p$ needed for successful recovery is hard to gauge. To ensure a successful recover, these works require either 1) a small $\mu_0$ or 2) really large $m$ or $n$ (virtually infinite data), which are both not practical. This is what motivated our unconventional approach of converting matrix completion problems into matrix sensing problems, to get rid of the requirements on $p$. However, *we are not implying that our approach could solve MC with arbitrary $p$ to arbitrary accuracy*. In Theorem 2.2, we show how smaller $p$ leads to worse solution accuracy and sometimes vacuous bounds.

Additionally, we introduce some other popular techniques for matrix completion without explicit recovery guarantees, including greedy algorithms (Lee and Bresler, 2010; Wang et al., 2014), alternating minimization (Haldar and Hernando, 2009; Tanner and Wei, 2016; Wen et al., 2012), iteratively reweighted least squares(IRLS) methods (Mohan and Fazel, 2012; Radhakrishnan et al., 2025), iterative thresholding Klopp (2015) and Riemannian optimization (Mishra et al., 2014; Dai and Milenkovic, 2010)—reviewed comprehensively in Nguyen et al. (2019). Another interesting line of work converts inductive matrix completion into regular MC (Ghassemi et al., 2018; Zilber and Nadler, 2022b), but they have extra information matrices $A, B$ not considered in (2), so it's hard to compare this work to the classic setting.

POWER OF OVER-PARAMETRIZATION

Recent studies have highlighted over-parametrization as a crucial strategy in matrix sensing when RIP constants $\delta_{2r}$ are suboptimal (i.e., $\delta_{2r} \geq 1/2$). Research by Zhang (2021; 2022) examined cases where the search rank $r_{\text{search}}$ exceeds the true rank $r$, thus increasing the problem's parametrization. Zhang (2022) demonstrated that for $r_{\text{search}} > r[(1 + \delta_n)/(1 - \delta_n) - 1]^2/4$ and $r \leq r_{\text{search}} < n$, each solution $\hat{X}$ satisfies $\hat{X}\hat{X}^\top = M^*$. Similarly, Ma and Fattahi (2022) established analogous results under RIP-type conditions for the $\ell_1$ loss. In addition to the classic over-parametrization of increasing $r_{\text{search}}$, convex relaxation (Recht et al., 2010) is certainly one popular approach by increasing parameters from $\mathcal{O}(nr)$ to $\mathcal{O}(n^2)$. Yalcin et al. (2023) showed that the RIP threshold for exact recovery using SDP can approach 1 when $M^*$ has a high true rank, thus underscoring the efficacy of over-parametrization. Nevertheless, the practical applicability of these conditions is limited, leading Ma et al. (2023) to explore an alternative approach to over-parametrization by lifting the search space of (1) into general tensor space and bank on important concepts from Sums-of-Squares optimization (Parrilo, 2003; Lasserre, 2001) to convert spurious local minimizers of (1) into strict saddle points in the lifted space so that they could be escaped by modern optimizers. Despite its utility in resolving spurious solutions, this tensor approach's applicability to matrix completion remains constrained by the need for a valid RIP constant.

## 1.2 NOTATION

Scalar values such as $\sigma_i(M)$ and $\lambda_i(M)$ represent the $i$-th largest singular value and eigenvalue of matrix $M$, respectively. The inner product between two matrices $\langle A, B \rangle$ is defined as $\mathrm{tr}(A^\top B)$. The Euclidean norm of a vector $v$ is denoted as $\|v\|$, while $\|M\|_F$ and $\|M\|_2$ are used for the Frobenius and induced $l_2$ norms of a matrix $M$. Vectorization $\mathrm{vec}(M)$ stacks the columns of $M$ into a vector, where $\mathrm{mat}(v)$ is its reserve operation. The set of integers from 1 to $n$ is expressed as $[n]$. Moreover, $\circ l$ indicates a repeated Cartesian product for $l$ times, $\oslash$ refers to the Kronecker product, and $\otimes$ signifies the tensor outer product. If these notations come with subscripts, they denote the dimension along which the operation is performed. Finally, if $S \in [n] \times [m]$ represents a subset of indices of a $n \times m$ matrix, then $N_S$ refers to the sub-matrix of $N \in \mathbb{R}^{n \times m}$ relevant to $S$ as per (3), and $\|N\|_{S,F}$ denotes the Frobenius norm of $N_S$.

## 2 THE PERTURBED MATRIX COMPLETION FORMULATION

As explained above, most literature regarding the recovery guarantees of matrix sensing problems require some valid RIP (RSC and RSS) constant. However, the attainment of such a constant automatically implies a trivial nullspace, meaning that $\mathcal{A}$ only maps a zero matrix to a zero vector, which is impossible for matrix completion problems. To demonstrate why this is, let's consider a $2 \times 2$ matrix recovery problem, and say we observed three entries of some $M \in \mathbb{R}^{2 \times 2}$ except for the lower-right entry. This will correspond to the case where

$$\mathcal{A}_\Omega(M) = \mathrm{vec}\left(\begin{bmatrix} 1 & 1 \\ 1 & 0 \end{bmatrix} \odot M\right) = \begin{bmatrix} 1 & 0 & 0 & 0 \\ 0 & 1 & 0 & 0 \\ 0 & 0 & 1 & 0 \\ 0 & 0 & 0 & 0 \end{bmatrix} \mathrm{vec}(M) := T_\Omega \, \mathrm{vec}(M) \in \mathbb{R}^4 \qquad (4)$$

For this $\mathcal{A}_\Omega$ to exhibit any RIP constant $\delta < 1$, it is required that $\|\mathcal{A}_\Omega(M)\|_2^2 \geq (1 - \delta)\|M\|_F^2$, meaning that the $T_\Omega$ matrix above cannot output $0$ unless $M$ is a zero matrix. Nevertheless, for this specific example, even if we observed three out of four entries, we can simply set $\mathrm{vec}(M) = [0, 0, 0, 1]^\top$ to make $\mathcal{A}_\Omega(M) = 0$, violating the RIP condition. This is a simple example showing us that RIP condition will not hold for matrix completion problems unless all entries are observed, regardless of its size. Therefore, it begs the question of *whether we could use the better studied, more powerful over-parametrized MS framework to offer guarantees for MC problems?* In this work, we give a positive answer to this question via the following three steps:

1. **Surrogate Construction:** We construct a surrogate problem ($\epsilon$-MC) to solve by slightly changing $\mathcal{A}_\Omega$. $\epsilon$-MC problem is a noisy MS problem that has more favorable theoretical properties.
2. **Surrogate Accuracy:** We prove that the surrogate's global solution approximates the true matrix $M^*$ with high probability under mild conditions.
3. **Tensor Framework Adaptation:** To address high RIP constants in the perturbed MS setting, we extend and adapt the lifted tensor framework from Ma et al. (2024), leveraging over-parametrization to achieve recovery.

Despite the lack of RIP constant, a surprisingly simple solution exists. The primary issue is the zero entries in the diagonal of $T_\Omega$, contributing to a non-trivial nullspace. By perturbing these zero entries slightly with a small number $\epsilon \in (0, 1]$, we can eliminate the nullspace. Revisiting (4), consider a perturbed sensing operator $\mathcal{A}_{\Omega,\epsilon}$:

$$\mathcal{A}_{\Omega,\epsilon}(M) = \mathrm{vec}\left(\begin{bmatrix} 1 & 1 \\ 1 & \epsilon \end{bmatrix} \odot M\right) = \begin{bmatrix} 1 & 0 & 0 & 0 \\ 0 & 1 & 0 & 0 \\ 0 & 0 & 1 & 0 \\ 0 & 0 & 0 & \epsilon \end{bmatrix} \mathrm{vec}(M) := T_{\Omega,\epsilon} \, \mathrm{vec}(M) \qquad (5)$$

in which $T_{\Omega,\epsilon}$ has a trivial nullspace as promised. However, different operators lead to different observations. For example, when considering the case of (4), using $\mathcal{A}_\Omega$ and $\mathcal{A}_{\Omega,\epsilon}$ on the same matrix results in:

$$\mathcal{A}_\Omega(M) = [M_{1,1} M_{1,2} M_{2,1} 0]^\top \longrightarrow \mathcal{A}_{\Omega,\epsilon}(M) = [M_{1,1} M_{1,2} M_{2,1} \epsilon M_{2,2}]^\top$$

Therefore, the original observation $b$ can be considered a noisy observation under the context of $\mathcal{A}_{\Omega,\epsilon}$, with

$$b = \mathcal{A}_{\Omega,\epsilon}(M^*) + w_\epsilon, \quad w_\epsilon = \begin{bmatrix} 0 & 0 & 0 & -\epsilon M_{2,2}^* \end{bmatrix}^\top \qquad (6)$$

where $w_\epsilon \in \mathbb{R}^{n^2}$ can be considered a noise term. With this idea in place, we formally introduce our perturbed MC problem to solve:

$$\min_{X \in \mathbb{R}^{n \times r_{\text{search}}}} \|\mathcal{A}_{\Omega,\epsilon}(XX^T) - b\|_2^2 := f_{w_\epsilon}(XX^\top) \quad (\epsilon\text{-MC}) \tag{7}$$

Essentially, we're transforming a noiseless matrix completion problem into a noisy matrix sensing problem with operator $\mathcal{A}_{\Omega,\epsilon}$ and deterministic noise $w_\epsilon$. This approach, notably, ensures the attainment of valid RSS/RSC parameters, equivalent to RIP constants.

**Lemma 2.1.** *Given an arbitrary matrix completion problem with sensing operator $\mathcal{A}_\Omega$, if this operator is perturbed to produce $\mathcal{A}_{\Omega,\epsilon}$ according to (5) with a scalar $\epsilon \in (0, 1]$, then the $\epsilon$-MC problem will exhibit $(1, n)$-RSS property and the $(\epsilon^2, n)$-RSC property.*

The above RSC constant is tight if it needs to hold for all possible MC instances, but could be drastically improved if we only consider specific formulations. We have included a generalized version of this Lemma (and its proof) that may serve independent interests outside this work, presented as Lemma D.1 in Appendix D. With that said, another major challenge that the perturbed formulation of $\epsilon$-MC problems brings is that the global solution of $\epsilon$-MC might not be $M^*$ anymore. This can be easily seen since $\|\mathcal{A}_{\Omega,\epsilon}(M^*) - b\|_2^2 \neq 0$. In other words, since the core idea of our approach is to solve the surrogate $\epsilon$-MC problem in order recover $M^*$, we need to know the conditions under which $M^*$ will be close to the global solution of (7), since later on we will show that over-parametrization via lifting could help us reach the global solution, denoted as $M^\dagger$, with guarantees. Inspired by Ma and Fattahi (2023), we hope to link it with the number of corrupted observations. If we adopt the standard assumption that each entry of the matrix is independently observed with probability $p$, then we could generalize this observation by linking it to $p$. As our next step, we show that $M^\dagger$ will be very close to $M^*$ with high probability, and we can further achieve a tradeoff between sample rate $p$ and geometric uniformity captured by $\epsilon$.

We will briefly go over the high-level ideas in this derivation and present our formal theorem in the end. Since we assumed that $M^\dagger$ is the global optimum of (7), then by definition it gives that $f_{w_\epsilon}(M^\dagger) - f_{w_\epsilon}(M^*) \leq 0$. If we partition the set $\Omega$ into $\bar{S}$, the observed, noiseless entries, and $S$, the unobserved, perturbed entries, then we could decompose $f_{w_\epsilon}(M^\dagger) - f_{w_\epsilon}(M^*)$ as:

$$
\begin{aligned}
0 \geq f_{w_\epsilon}(M^\dagger) - f_{w_\epsilon}(M^*) =& \frac{1}{2}\|\mathcal{A}_{\Omega,\epsilon}(M^\dagger - M^*)\|_{\bar{S},2}^2 + \frac{1}{2}\|\mathcal{A}_{\Omega,\epsilon}(M^\dagger - M^*) - w_\epsilon\|_{S,2}^2 - \frac{1}{2}\|w_\epsilon\|_{S,2}^2 \\
=& \frac{1}{2}\|\mathcal{A}_{\Omega,\epsilon}(M^\dagger - M^*)\|_{\bar{S},2}^2 + \frac{1}{2}\|\mathcal{A}_{\Omega,\epsilon}(M^\dagger)\|_{S,2}^2 - \frac{1}{2}\epsilon^2\|M^*\|_{S,F}^2 \\
\geq& \frac{1}{2}\|\mathcal{A}_{\Omega,\epsilon}(M^\dagger - M^*)\|_{\bar{S},2}^2 - \frac{1}{2}\epsilon^2\|M^*\|_{S,F}^2
\end{aligned}
$$
$$\tag{8}$$

where $\|\cdot\|_{S,2}$ denotes the $l_2$ norm of the sub-vector with entries in set $S$. Then if we add $\frac{1}{2}\|M^\dagger - M^*\|_{S,2}^2$ to both sides of (8), it is easy to show

$$\frac{1}{2}\|M^\dagger - M^*\|_F^2 \leq \frac{1}{2}\left(\|M^\dagger - M^*\|_{S,F}^2 + \epsilon^2\|M^*\|_{S,F}^2\right) \tag{9}$$

Here we look into the right hand side terms of (9) a bit more carefully, and realize that both $\|M^\dagger - M^*\|_{S,F}^2$ and $\|M^*\|_{S,F}^2$ are random variables with their sizes dependent on the sampling rate. Since $\|\cdot\|_S^2$ only denotes the size of the sub-matrix that are not observed (therefore perturbed by $\epsilon$), if our sample rate $p$ is large, this norm would also be small in expectation. By further using concentration inequality to control deviation, we show why the $\epsilon$-MC problem (7) can serve as a meaningful surrogate to the original MC problem in our main theorem:

**Theorem 2.2.** *Assume that $M^\dagger \in \mathbb{R}^{n \times n}$ is a symmetric, rank-$r$ matrix that is a global optimum of (7) with an $\epsilon \in (0, 1]$. Assume that each entry of the original MC problem is independently observed with probability $p$, then for any $\chi \leq p \in \mathbb{R}$,*

$$\|M^\dagger - M^*\|_F^2 \leq \frac{1 - p + \chi}{p - \chi}\epsilon^2\|M^*\|_F^2 \tag{10}$$

*holds with probability at least $1 - \exp\left(-2\chi^2\|d\|_1^2/\|d\|_2^2\right)$, where $d \in \mathbb{R}^{n^2}$ is defined as*

$$d := \text{vec}(M^\dagger - M^*) \odot \text{vec}(M^\dagger - M^*) + \epsilon^2 \text{vec}(M^*) \odot \text{vec}(M^*) \tag{11}$$

We begin by noting that for any vector $d$, elementary inequalities ensure that $1 \leq \|d\|_1^2/\|d\|_2^2 \leq n^2$, and this ratio increases as the entries of $d$ become more evenly distributed. The proof of this theorem is given in Appendix D; we employ Hoeffding's inequality to obtain a clean, interpretable form. Other concentration tools (e.g., Bennett's inequality) can also be applied, but we chose Hoeffding's for its simplicity. **Conceptually, Theorem 2.2 departs from exact-recovery guarantees and instead provides an explicit accuracy–probability trade-off that holds for any** $p \in (0,1]$**: the *solution accuracy of $\epsilon$-MC depends smoothly on $p$ and the user-chosen $\epsilon$.** When $p$ is small, $\chi$ must also be small regardless of $\epsilon$, so the probability for (10) to hold can be low; when $p$ is larger, the left-hand side of (10) can be made small without taking $\epsilon$ to be tiny, enabling lower computational demand (see Section 4). For context, information-theoretic lower bounds for *unique* recovery of $M^*$ imply that $p$ must exceed instance-dependent thresholds (e.g., Pimentel-Alarcón et al. (2016)). In contrast, our goal is not perfect recovery of $M^*$ but to find $M^\dagger$ that is provably close to $M^*$; this relaxes sampling requirements and supports the use of over-parameterization to obtain meaningful guarantees even in low-observation regimes.

Figure 1 visualizes this result with 2D heat maps. This figure shows, for each target bound on $\|M^\dagger - M^*\|_F^2$ (x-axis) and each $\epsilon$ (y-axis), the smallest probability lower bound delivered by (10). We display curves for $p = 0.3, 0.4, 0.5$. As $p$ increases, similar accuracies are certified with higher probability (brighter color). For instance, let's consider the $p = 0.3$ case. Figure 1 shows $\|M^\dagger - M^*\|_F^2 \leq 0.02$ holds with probability at least $0.8$ over a wide range of $\epsilon$ values. By contrast, invoking Theorem 3 of Pimentel-Alarcón et al. (2016) shows that, for a $50 \times 50$ matrix, at least 29 observed entries per column are required to achieve the same $0.8$ probability of *unique* recovery of $M^*$—which is highly unlikely when $p = 0.3$. This highlights the practical advantage of our $\epsilon$-MC approach in regimes where exact recovery is information-theoretically out of reach.

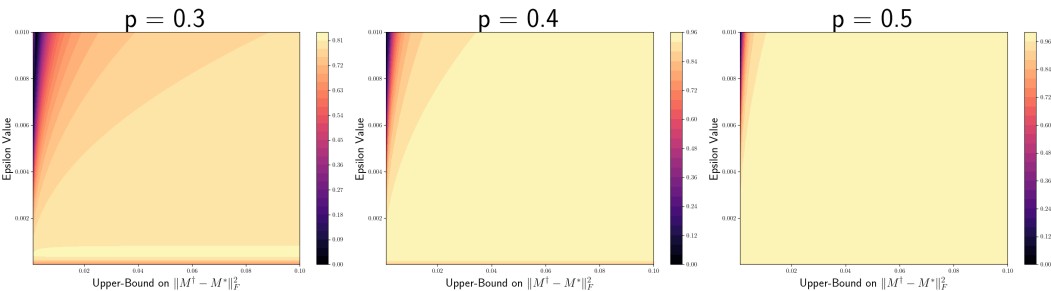

Figure 1: Probability Lower-Bound for Theorem 2.2.

## 3 LIFTED TENSOR FRAMEWORK WITH NOISE

Now that we are able to reformulate the original MC problem (2) into the new $\epsilon$-MC problem (7), it presents us with a new challenge. Although now (7) admits valid RSC/RSS constants, this is nevertheless still a difficult matrix sensing problem to solve due to its small RSC constant (or large RIP constant). Thus, it is important that we apply an over-parametrized framework to deal with it in order to compensate for the poor geometric uniformity.

To this end, we employ the lifted tensor framework proposed in Ma et al. (2023), since it has the ability to deal with really small RSC constants, like those we have in $\epsilon$-MC. However, in their original work, measurements were assumed to be clean, and this is incompatible with our framework since we hope to deal with noisy MS problems. Thus, we generalize the original results in Ma et al. (2023), and also in its subsequent work Ma et al. (2024) to demonstrate how the inclusion of noise could affect guarantees in when using a higher-order tensor parametrization.

First of all, we present the lifted tensor problem when our observations are corrupted by some random noise $\tilde{w} \in \mathbb{R}^m$,

$$\min_{\mathbf{w} \in \mathbb{R}^{n r o l}} \quad \|\langle \mathbf{A}^{\otimes l}, \langle \mathbf{P}(\mathbf{w}), \mathbf{P}(\mathbf{w})\rangle_{2*[l]}\rangle - \tilde{b}^{\otimes l}\|_F^2 \tag{12}$$

where $\tilde{b} = \mathcal{A}(M^*) + \tilde{w}$, and $\mathbf{A} \in \mathbb{R}^{m \times n \times n}$ is a three-way tensor which can be seen as a concatenation of all sensing matrices $\{A_i\}_{i=1}^m$, and $\mathbf{w} \in \mathbb{R}^{nrol}$ is the tensor decision variable used to increase the

parametrization of $X \in \mathbb{R}^{n \times r}$. Here, $^{\otimes l}$ simply denotes $l$ times of repeated tensor outer product, and $\mathbf{P}$ is just another constant permutation tensor used for correct multiplication. The gist of this paper is not on the tensor formulation, thus many details are deferred to Appendix C, and interested readers can learn more about general tensor knowledge and problem details there. For convenience's sake, we define $f^l(\cdot) : \mathbb{R}^{n \circ 2l} \mapsto \mathbb{R}$ and $h^l(\cdot) : \mathbb{R}^{[n \times r] \circ l} \mapsto \mathbb{R}$ as $f^l(\mathbf{M}) := \|\langle \mathbf{A}^{\otimes l}, \mathbf{M}\rangle - \tilde{b}^{\otimes l}\|_F^2$ and $h^l(\mathbf{w}) = f^l(\langle \mathbf{w}, \mathbf{w}\rangle_{2*[l]})$, with $\nabla f^l(\cdot) = \nabla_{\mathbf{M}} f^l(\cdot)$ and $\nabla h^l(\cdot) = \nabla_{\mathbf{w}} h^l(\cdot)$.

In the original works, it was proven that the lifted formulation (12) is able to convert spurious second-order points in (1) to strict saddle points (critical points with local escape direction) via its drastic over-parametrization if this spurious solution is somehow far away from the ground truth $M^*$. However, the first thing to note here is that for any corrupted MS problem (its observation $b$ is not clean and affected by noise), its global solution might not correspond to $M^*$ anymore, which is the same challenge that we faced in the $\epsilon$-MC formulation. This means that spurious solutions have to be even more distant to $M^*$ for it be converted into strict saddles, depending on the intensity of noise. The result is summarized in the following theorem:

**Theorem 3.1.** *Consider an arbitrary second-order point $\hat{X} \in \mathbb{R}^{n \times r}$ of the factorized matrix sensing objective in the form of* (1) *where its observations $b$ could be potentially corrupted by some random noise $\tilde{w} \in \mathbb{R}^m$ (i.e. $b = \tilde{b}$). Assuming that the linear operator $\mathcal{A}(\cdot)$ in* (1) *satisfies the RSC and RSS conditions with constants $\alpha_s, L_s$ respectively. Then $\hat{\mathbf{w}} = \text{vec}(\hat{X})^{\otimes l}$ is a strict saddle of* (12) *with a rank-1 symmetric escape direction if*

$$\|M^* - \hat{X}\hat{X}^\top\|_F^2 \geq \frac{L_s}{\alpha_s}\lambda_r(\hat{X}\hat{X}^\top)\operatorname{tr}(M^*) + \frac{\|\tilde{w}\|_2^2}{\alpha_s} \tag{13}$$

*with an odd $l$ satisfying*

$$l > \frac{1}{1 - \log_2(2\beta)}, \quad \beta := \frac{L_s \operatorname{tr}(M^*)\lambda_r(\hat{X}\hat{X}^\top)}{\alpha_s\|M^* - \hat{X}\hat{X}^\top\|_F^2 - \|\tilde{w}\|_2^2}. \tag{14}$$

The proof of this theorem is located in Appendix D. The theorem highlights how the conversion radius from spurious solutions to strict saddles is influenced by the norm of the noise $\tilde{w}$. Setting $\tilde{w} = 0$ allows this theorem to coincide with Theorem 4 from Ma et al. (2024). More importantly, it is crucial for the critical point $\hat{\mathbf{w}}$ in (12) to be a rank-1 tensor to possess a negative escape direction. For a detailed definition of tensor rank, please see Appendix C. According to Ma et al. (2024), employing a gradient descent (GD) algorithm with sufficiently small initialization ensures that the search is conducted over approximately rank-1 tensors throughout the GD trajectory. This work further establishes that this characteristic remains unchanged when $b$ is substituted with $\tilde{b}$, showing that the effects of implicit bias induced by vanilla GD is agnostic of noise. Here we present an informal version of this result to facilitate understanding, and the full version and its proof can be found in Appendix C and Appendix D respectively:

**Theorem 3.2** (Informal)**.** *Consider a finite-horizon gradient descent trajectory $\{\mathbf{w}_t\}_{t \in [T]}$ of* (12) *with $\mathbf{w}_{t+1} = \mathbf{w}_t - \eta\nabla h^l(\mathbf{w}_t)$ starting from the initialization $\mathbf{w}_0 = \xi x_0^{\otimes l}$ with $\xi \in \mathbb{R}$ denoting the scale of the initialization, $\eta$ representing the step-size and $x_0 \in \mathbb{R}^{nr}$ being an arbitrary vector with $\|x_0\|_2^2 = 1$. Then for a sufficiently small $\xi$, there exists an iteration number $t(\kappa, l) \geq 1$ that depends on an arbitrary constant $\kappa < 1$ and lifting degree $l$ such that*

$$\frac{\lambda_2^v(\mathbf{w}_t)}{\lambda_1^v(\mathbf{w}_t)} \leq \kappa, \quad \forall t \in [t(\kappa, l), T] \tag{15}$$

*where $\lambda_i^v(\cdot)$ denotes the $i$-th largest v-eigenvalue (see Appendix C) of a given tensor, a min-max definition of tensor eigenvalues that serve optimization purposes well. This implies that $\mathbf{w}_t$ will be approximately rank-1 after iteration $t(\kappa, l)$ along the GD trajectory $\{\mathbf{w}_t\}_{t \in [T]}$. Furthermore, $t(\kappa, l)$ increases with a smaller $\kappa$, meaning that the tensor along the trajectory will become increasingly like rank-1 as GD updates happen.*

The main takeaway is that by incorporating the noise $\tilde{w}$ into our objective (12), the ability of gradient descent algorithms to induce implicit bias (implicitly penalizing towards compact representations, like rank-1 tensors) remains unchanged. It is also worth noting that the results presented in this subsection apply to all tensor problems in the form of (12), which are lifted from general noisy matrix sensing problems, and not specific to our $\epsilon$-MC problem.

## 4 MAIN RESULTS

Our goal is to find a globally optimal solution for the $\epsilon$-MC problem because it closely represents $M^*$ via Theorem 2.2. However, this becomes challenging due to the underlying $\alpha_s$ of (7), which depends on the small value of $\epsilon$. To address this, rather than solving the problem using its basic matrix factorized form (as shown in equation (1)), which lacks global optimization guarantees, we apply the tensor formulation (12). This framework has the ability to verifiably extract (see Theorem 3.1) the global solution up to a certain accuracy via increasing parametrization.

By combining Theorems 2.2 and 3.1, we demonstrate a new way to approximately solve the generic MC problem (as formulated in equation (2)) while still providing reliable global solutions, as elaborated in our main theorem below

**Theorem 4.1.** *Consider the matrix completion problem of completing a $n \times n$, rank-$r$ matrix $M^*$, where $\Omega \subseteq [n] \times [n]$ denotes the set of observed entries and $\bar{\Omega}$ denotes the unobserved entries. Introduce a perturbation $\epsilon \in (0, 1]$ to formulate an $\epsilon$-MC problem as per (7). Applying the tensor framework described in (12) to this $\epsilon$-MC problem yields the following results:*

*For any rank-1 critical point $\hat{\mathbf{w}} = \text{vec}(\hat{X})^{\otimes l}$ of (12), if it is a second-order point (local minima), this implies that*

$$\|M^* - \hat{X}\hat{X}^\top\|_F < \frac{1}{\epsilon}\lambda_r(\hat{X})\sqrt{\text{tr}(M^*)} + e_1 \tag{16}$$

*holds with probability at least $q$, under the condition that $l$ is odd and meets the requirement:*

$$l > \frac{1}{1 - \log_2(2\beta)}, \quad \beta := \frac{\text{tr}(M^*)\lambda_r(\hat{X}\hat{X}^\top)}{\epsilon^2\left(\|M^* - \hat{X}\hat{X}^\top\|_F^2 - e_2\right)}. \tag{17}$$

*For all instances of the MC problem, the following hold:*

$$e_1 = e_2 = \|M^*\|_{\bar{\Omega}, F}^2, \quad q = 1 \tag{18}$$

*Alternatively, if all entries are observed independently with probability $p$ (a special case in which Theorem 2.2 can be used), $e_2$ could be ignored (i.e. $e_2 = 0$) and:*

$$e_1 = \sqrt{\frac{1 - p + \chi}{p - \chi}}\epsilon\|M^*\|_F, \; q = 1 - e^{\left(-2\chi^2\|d\|_1^2/\|d\|_2^2\right)} \tag{19}$$

*where $\chi$ and $d$ are defined as per Theorem 2.2.*

Our main theorem builds on the results of Theorem 3.1, applying specific parameters ($L_s = 1$, $\alpha_s = \epsilon^2$) along with the definition of $w_\epsilon$ from equation (6). This leads to a deterministic outcome where the probability $q$ equals 1. However, there's a critical aspect to consider: the transition from a spurious solution is contingent upon the condition described in (16). A significant challenge arises if $\|M^*\|_{\bar{\Omega}, F}^2$ is large, potentially rendering this bound vacuous. Here, the utility of Theorem 2.2 becomes evident. Under its probabilistic framework, we apply a triangle inequality to reduce the bound $e_1$ to $\sqrt{\frac{1-p+\chi}{p-\chi}}\epsilon\|M^*\|_F$. This simplification is critical, as the presence of $\epsilon$ and $p$ can significantly minimize the error term, countering the inaccuracies introduced by the $\epsilon$-MC framework. Also note that although the bound (16) contains the term $\text{tr}(M^*)$, which we previously said was unaccessible (especially in the incoherence calculation), knowing it or not in advance will not affect whether the problem could be solved using the lifted framework, and it only affects the theoretical guarantees describing worst-case scenarios. The proof to this theorem can be found in Appendix D.

Therefore, $\epsilon$ now is an user-defined parameter that balances solution accuracy (through $e_1$) and the amount of parametrization needed (via equation (17)), and can be determined by user requirements. **In reality, tensor iterates $w_t$ on the gradient descent trajectory of (12) are approximate rank-1 tensors as detailed in Theorem 3.2, thus Theorem 4.1 cannot be used directly to characterize the landscape since it only applies to exactly rank-1 tensors. To this end, Theorem C.10, an extended version of Theorem 4.1, is used to actually describe the recovery of global optima under these more realistic conditions**. The two theorems only differ in some technical details, thus for simplicity's sake we defer Theorem C.10 to Appendix C to help readers understand the main results better.

## 5 NUMERICAL EXPERIMENTS

In this section, we evaluate our method[1] against both classic and state-of-the-art solvers on two families of matrix completion (MC) instances. In addition to the uniformly and independently sampled instances as in Theorem 2.2, we also include the hard instance from Yalçın et al. (2022), which is known to admit exponentially many spurious solutions:

$$\Omega = \{(i,i), (i, 2k), (2k, i) \mid \forall i \in [n], \ k \in [\lfloor n/2 \rfloor]\}. \tag{20}$$

We compare our methods to: (i) the classic Burer–Monteiro (BM) factorization baseline for (2); (ii) the SDP relaxation (Candès and Tao, 2010); (iii) GNMR (Zilber and Nadler, 2022a), a state-of-the-art BM method that solves a linearized least-squares subproblem each iteration (we use its Algorithm 3 with $\alpha = -1$); (iv) Lin–RFM (Radhakrishnan et al., 2025), an SVD-free IRLS/Schatten-$p$ spectral-reweighting scheme; and (v) our lifted approach (12) with $l = 3$. Our method produces a tensor $\mathbf{w}_T$ after $T$ gradient steps, and Tensor PCA (Ma et al., 2024) extracts a principal component $X_T \in \mathbb{R}^{n \times r}$ via $\mathbf{w}_T \approx \text{vec}(X_T)^{\otimes l}$, with $X_T$ being returned for (2). We declare success if $\|X_T X_T^\top - M^*\|_F \leq 0.05$ and report *success rates* over 20 trials per problem, rather than average reconstruction error, because they are robust to outliers and better capture instances where $\epsilon$-MC may be an invalid surrogate (per Theorem 2.2). Further experiments on effects of $\epsilon$ and $p$ are included in Appendix B.

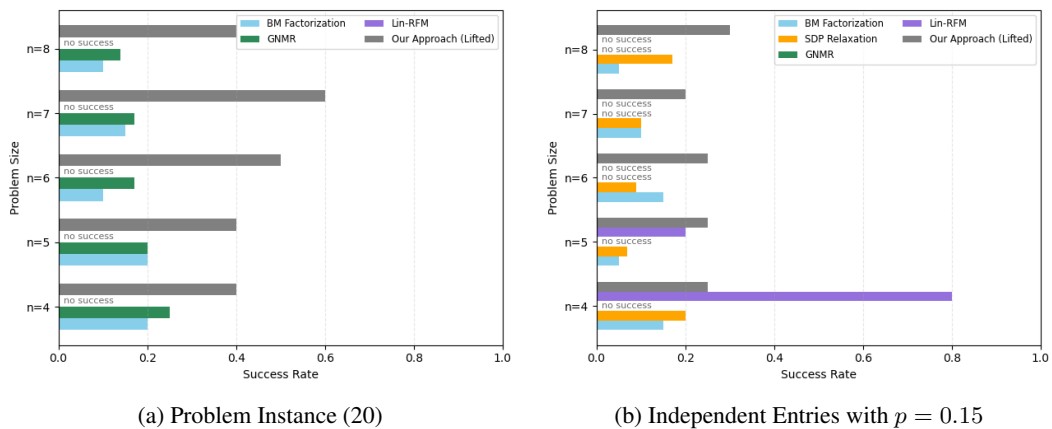

(a) Problem Instance (20)  (b) Independent Entries with $p = 0.15$

Figure 2: We use $\epsilon = 5 * 10^{-5}$, a learning rate of 2e-2, an initialization scale of $\xi = 10^{-4}$ for (12). SDP was not deployed for (a) since the solution is not unique in the relaxed regime. For (b), the variability of $\Omega$ necessitates running 10 distinct problem instances for each size $n$.

We see from figure 2 that across both instance families and sizes $n$, our lifted approach attains consistently higher success rates. GNMR is competitive on the structured instance but degrades on the independent model under the tested setting ($\alpha = -1$), while Lin–RFM, as an IRLS/Schatten-$p$ method, shows strong performance at smaller sizes in the independent model and is less effective on the structured instance. Since high-order tensors require substantial RAM/VRAM to store, we limit our experiments to small-scale settings given our current compute budget. We are actively exploring methods to leverage the tensor framework without incurring such high computational costs, particularly by investigating deterministic escape directions in the matrix space.

## 6 CONCLUSION

In conclusion, we shift from the classical goal of exact recovery of $M^*$ to solving a carefully designed surrogate that remains provably close to $M^*$ and whose landscape is amenable to over-parameterization. By converting MC into the $\epsilon$-MC sensing surrogate, we obtain explicit accuracy–probability guarantees as functions of the sampling rate $p$ and user-chosen $\epsilon$ (Theorem 2.2). Coupled with a lifted tensor parameterization, this yields verifiable routes to global solutions even with very few observations (Theorem 4.1), echoing the broader observation that larger models can make hard objectives tractable. We believe this also serves as a testament to why a certain degree of perturbation and randomness is indispensable in modern over-parametrized machine learning.

---

[1]Code in the supplementary material; experiments run on an M1 Max MacBook Pro.

REPRODUCIBILITY STATEMENT

This paper is fully committed to reproducibility. For the theoretical results, we clearly state all assumptions, provide complete proofs in Appendix D, and include step-by-step derivations with citations to relevant prior work where appropriate. For the computational results, all code is included in the supplementary material, organized into four files with well-structured functions. Each function is given descriptive names and intuitive variables to ensure clarity and facilitate reproducibility. We also included a dedicated function to show the visualization of Figure 1.

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

## A  LLM USE

This paper uses LLM exclusively for grammatical checks and improving flow of English. No LLM was used for math derivation nor the writing of code.

## B  ADDITIONAL ABLATION STUDIES

In this section we complement our main experiments with ablation studies on the user-chosen perturbation level $\epsilon$ and the sampling probability $p$ for the structured instance in problem (20). Throughout, we consider the setting $n = 6$, depth $\ell = 3$, and initialization magnitude $10^{-4}$.

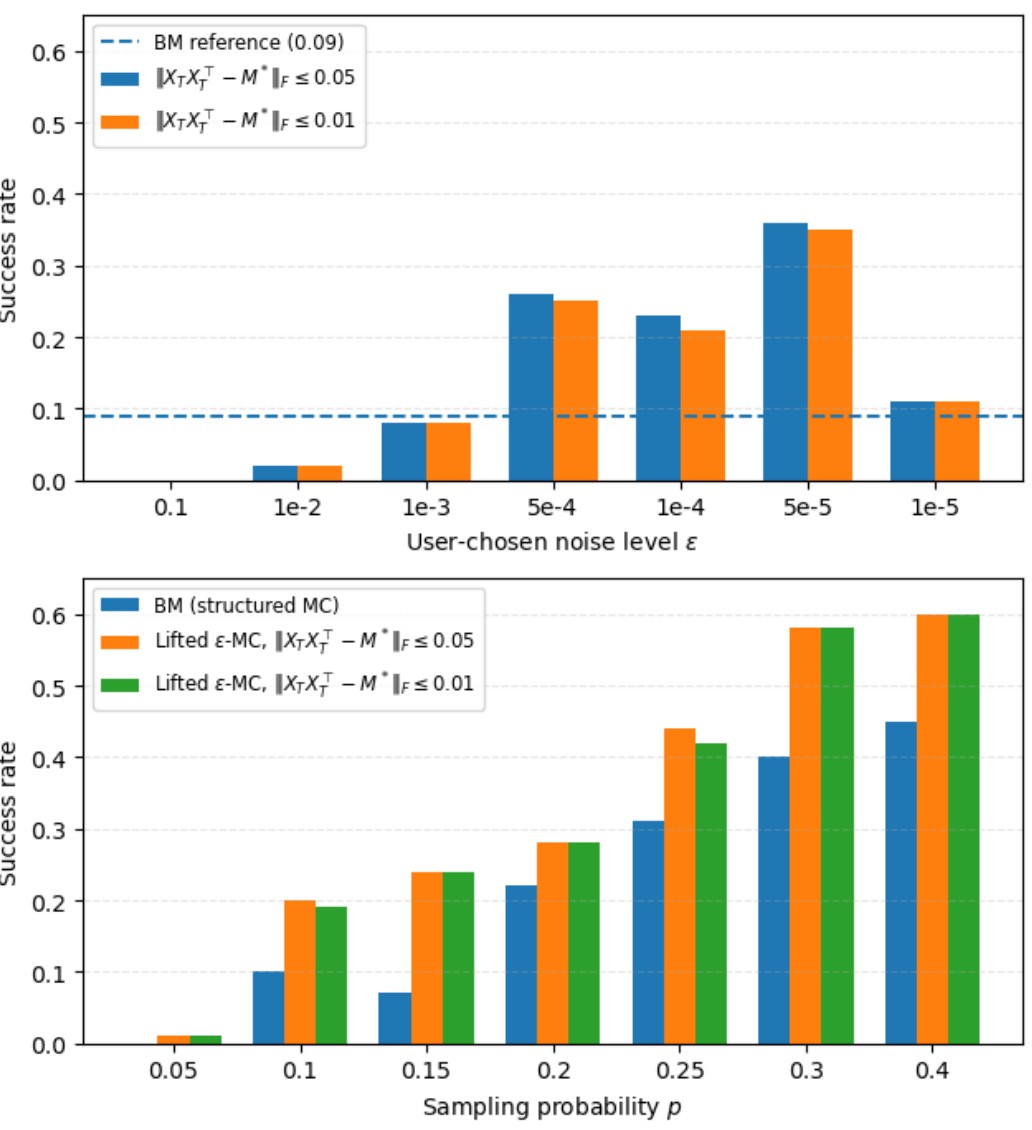

Figure 3: Ablation studies for the lifted $\epsilon$-MC formulation with $n = 6, \ell = 3$, initialization magnitude $10^{-4}$. *Top:* Success rates with respect to the perturbation level $\epsilon$, under two reconstruction criteria $\|X_T X_T^\top - M^\star\|_F \leq 0.05$ and $\|X_T X_T^\top - M^\star\|_F \leq 0.01$, together with the BM baseline success rate (dashed line) for the structured instance (20). Success rate is polled from 100 independent trials each. *Bottom:* Success rates of the independently sampled problem with respect to sampling probability $p$, comparing the BM approach and the lifted $\epsilon$-MC method under both reconstruction criteria. For each $p$ level, 10 random masks are generated and 20 trials each are run for a total of 200 trials.

**Ablation on $\epsilon$.** For the first study we fix the sampling pattern as in problem (20) and vary the perturbation level $\epsilon \in \{0.1, 10^{-2}, 10^{-3}, 5\times 10^{-4}, 10^{-4}, 5\times 10^{-5}, 10^{-5}\}$. For each $\epsilon$, we record the empirical success rates under the two reconstruction criteria $\|X_T X_T^\top - M^\star\|_F \leq 0.05$ and $\|X_T X_T^\top - M^\star\|_F \leq 0.01$. As shown in the top panel of Figure 3, the success curves for the two thresholds closely track each other, indicating that this problem behaves as a "success or failure" scenario rather than exhibiting a gradual degradation in reconstruction quality.

These trends are consistent with the error bound in Theorem 4.1 (cf. inequality (16)), which decomposes into a *computational part*, proportional to $1/\epsilon$ (this rate is rather conservative and could be improved, please see Lemma D.1), and an *accuracy part* $e_1$ capturing the mismatch between the $\epsilon$-perturbed objective and the original matrix completion problem. For large $\epsilon$ (e.g., 0.1 and $10^{-2}$), the perturbation error $e_1$ dominates and the success rate is low even if the optimizer reaches a global minimum of the $\epsilon$-MC landscape. As $\epsilon$ decreases into an intermediate regime (roughly between $10^{-3}$ and $5\times 10^{-5}$), the two terms balance and the success rate peaks. When $\epsilon$ becomes too small (e.g., $10^{-5}$), the $1/\epsilon$ factor renders the problem numerically more challenging, and the empirical success rate deteriorates.

**Ablation on sampling probability $p$.** In the second study we fix $\epsilon = 5\times 10^{-4}$ and vary the sampling probability $p \in \{0.05, 0.1, 0.15, 0.2, 0.25, 0.3, 0.4\}$. For each $p$, we again estimate the success rates over 10 randomly generated masks with 20 trials each amounting to 200 trials in total for both the BM method and our approach. The results are summarized in the bottom panel of Figure 3.

As $p$ increases, both methods become more successful, as expected from having more observed entries. The lifted $\epsilon$-MC approach exhibits a clear advantage in the low-$p$ regime, where the BM formulation struggles to recover the structured instance. At larger $p$, the BM baseline itself achieves high success rates, which naturally reduces the margin between the two methods. At the same time, further gains for our approach appear limited by the same computational term in the error bound: while the accuracy part improves rapidly with $p$, the difficulty of optimizing the lifted problem becomes the dominant factor.

Overall, these ablation studies illustrate how the choice of $\epsilon$ and the sampling probability $p$ jointly govern the empirical behavior of the lifted $\epsilon$-MC formulation.

## C  ADDITIONAL DETAILS FOR NOISY LIFTED FRAMEWORK

### C.1  ADDITIONAL DEFINITIONS

**Definition C.1** (RIP). (Candès and Recht, 2009) Given a natural number $p$, the linear map $\mathcal{A} : \mathbb{R}^{n \times n} \mapsto \mathbb{R}^m$ is said to satisfy $\delta_p$-RIP if there is a constant $\delta_p \in [0, 1)$ such that

$$(1 - \delta_p)\|M\|_F^2 \leq \|\mathcal{A}(M)\|^2 \leq (1 + \delta_p)\|M\|_F^2$$

holds for matrices $M \in \mathbb{R}^{n \times n}$ satisfying $\text{rank}(M) \leq p$.

**Definition C.2** (Tensor). As a generalization of the way vectors are used to parametrize finite-dimensional vector spaces, we use *arrays* to parametrize tensors generated from product of finite-dimensional vector spaces, as per Comon et al. (2008). In particular, we define an $l$-way array as such:

$$\mathbf{a} = \{a_{i_1 i_2 \dots i_l} | 1 \leq i_k \leq n_k, 1 \leq k \leq l\} \in \mathbb{R}^{n_1 \times \dots \times n_l}$$

Note that in this paper tensors and arrays can be regarded as synonymous since there exists an isomorphism between them. Moreover, if $n_1 = \dots = n_l$, then we call this tensor(array) an $l$-order(way), $n$-dimensional tensor. For the convenience of tensor representation, we use the notation $\mathbb{R}^{n \circ l}$ with $n \circ l := n \times \dots \times n$. In this work, tensors are denoted with bold variables, and other fonts are reserved for matrices, vectors, and scalars unless specified otherwise.

**Definition C.3** (Symmetric Tensor). Similar to the definition of symmetric matrices, for an order-$l$ tensor $\mathbf{a}$ with the same dimensions (i.e., $n_1 = \dots = n_l$), also called a cubic tensor, it is said that the tensor is symmetric if its entries are invariance under any permutation of their indices:

$$a_{i_{\sigma(1)} \cdots i_{\sigma(l)}} = a_{i_1 \cdots i_l} \quad \forall \sigma, \quad i_1, \dots, i_l \in \{1, \dots, n\}$$

where $\sigma \in \mathcal{G}_l$ denotes a specific permutation and $\mathcal{G}_l$ is the symmetric group of permutations on $\{1, \dots, l\}$. We denote the set of symmetric tensors as $\text{S}^l(\mathbb{R}^n)$.

**Definition C.4** (Rank of Tensors). The rank of a cubic tensor $\mathbf{a} \in \mathbb{R}^{n \circ l}$ is defined as

$$\text{rank}(\mathbf{a}) = \min\{r | \mathbf{a} = \sum_{i=1}^{r} u_i \otimes v_i \otimes \cdots \otimes w_i\}$$

for some vector $u_i, \ldots, w_i \in \mathbb{R}^n$. Furthermore, according to Kolda (2015), if $\mathbf{a}$ is a symmetric tensor, then it can be decomposed as:

$$\mathbf{a} = \sum_{i=1}^{r} \lambda_i u_i \otimes \cdots \otimes u_i \coloneqq \sum_{i=1}^{r} \lambda_i u_i^{\otimes l}$$

and the rank is conveniently defined as the number of nonzero $\lambda_i$'s, which is very similar to the rank of symmetric matrices indeed. The most important concept in our paper is rank-1 tensors, and for any tensor $\mathbf{a}$, a necessary and sufficient condition for it to be rank-1 is that

$$\mathbf{a} = u^{\otimes l}$$

for some $u \in \mathbb{R}^n$.

**Definition C.5** (Tensor Multiplication). Outer product is an operation carried out on a pair of tensors, denoted as $\otimes$. The outer product of 2 tensors $\mathbf{a}$ and $\mathbf{b}$, respectively of orders $l$ and $p$, is a tensor of order $l + p$, denoted as $\mathbf{c} = \mathbf{a} \otimes \mathbf{b}$ such that:

$$c_{i_1 \ldots i_l j_1 \ldots j_p} = a_{i_1 \ldots i_l} b_{j_1 \ldots j_p}$$

When the 2 tensors are of the same dimension, this product is such that $\otimes : \mathbb{R}^{n \circ l} \times \mathbb{R}^{n \circ p} \mapsto \mathbb{R}^{n \circ (l+p)}$. Henceforth, we use the shorthand notation

$$\underbrace{a \otimes \cdots \otimes a}_{l \text{ times}} \coloneqq a^{\otimes l}$$

We also define an inner product of two tensors. The mode-$q$ inner product between the 2 aforementioned tensors having the same $q$-th dimension is denoted as $\langle \mathbf{a}, \mathbf{b} \rangle_q$. Without loss of generality, assume that $q = 1$ and

$$[\langle \mathbf{a}, \mathbf{b} \rangle_q]_{i_2 \ldots i_l j_2 \ldots j_p} = \sum_{\alpha=1}^{n_q} a_{\alpha i_2 \ldots i_l} b_{\alpha j_2 \ldots j_p}$$

Note that when we write $\langle \cdot, \cdot \rangle_q$, we count the $q$-th dimension of the first entry. Indeed, this definition of inner product can also be trivially extended to multi-mode inner products by just summing over all modes, denoted as $\langle \mathbf{a}, \mathbf{b} \rangle_{q,\ldots,s}$.

**Lemma C.6** (Section 10.2 Petersen et al. (2008)). *For four arbitrary matrices $A, B, C, D$ of compatible dimensions, it holds that*

$$\langle A \otimes B, C \otimes D \rangle_{2,4} = AC \otimes BD \tag{21}$$

**Definition C.7** (Variational Eigenvalue of Tensors Ma et al. (2024)). For a given tensor $\mathbf{w} \in \mathbb{R}^{n \circ l}$, we define its $k^{th}$ variational eigenvalue (v-Eigenvalue) $\lambda_k^v(\mathbf{w})$ as

$$\lambda_k^v(\mathbf{w}) \coloneqq \max_{\substack{S \\ \dim(S)=k}} \min_{\mathbf{u} \in S} \frac{|\langle \mathbf{w}, \mathbf{u} \rangle|}{\|\mathbf{u}\|_F^2}, \quad k \in [n]$$

where $S$ is a subspace of $\mathbb{R}^{n \circ l}$ that is spanned by a set of orthogonal, symmetric, rank-1 tensors. Its dimension denotes the number of orthogonal tensors that span this space.

## C.2 FORMULATION DETAILS

As noted in our main formulation (12), the decision variable $\mathbf{w}$ is a tensor of dimension $nr \times \cdots \times nr$, since it serves as a repeated outer product of $\text{vec}(X)$ with $X \in \mathbb{R}^{n \times r}$ being our original decision variable in (1) (here we assume $r_{\text{search}} = r$). The permutation $\mathbf{P}$ is needed in order to convert $\mathbf{w} \in \mathbb{R}^{nr \circ l}$ back to $\mathbb{R}^{[n \times r] \circ l}$ in order to do meaningful inner products. $\mathbf{P} \in \mathbb{R}^{n \times r \times nr}$ is defined as

$$\langle \mathbf{P}, \text{vec}(X) \rangle_3 = X \quad \forall X \in \mathbb{R}^{n \times r}, n, r \in \mathbb{Z}^+$$

Such $\mathbf{P}$ can be easily constructed via filling appropriate scalar "1"s in the tensor. Via Lemma C.1, we also know that

$$\langle \mathbf{P}^{\otimes l}, \text{vec}(X)^{\otimes l}\rangle_{3*[l]} = (\langle \mathbf{P}, \text{vec}(X)\rangle_3)^{\otimes l} = X^{\otimes l} \tag{22}$$

Notationally, we abbreviate $\langle \mathbf{P}^{\otimes l}, \mathbf{w}\rangle_{3*[l]}$ as $\mathbf{P}(\mathbf{w})$ for enhanced readability for an arbitrary tensor $\mathbf{w}$ with dimension greater or equal to 2.

Since we also make extensive use of first and second order critical points of (12), we present them here for accessibility:

**Lemma C.8.** *The tensor $\hat{\mathbf{w}} \in \mathbb{R}^{nr\circ l}$ is an SOP of* (12) *if and only if*

$$\langle \nabla f^l(\langle \mathbf{P}(\hat{\mathbf{w}}), \mathbf{P}(\hat{\mathbf{w}})\rangle_{2*[l]}), \mathbf{P}(\hat{\mathbf{w}})\rangle_{2*[l]} = 0, \tag{23a}$$

$$2\langle \nabla f^l(\langle \mathbf{P}(\hat{\mathbf{w}}), \mathbf{P}(\hat{\mathbf{w}})\rangle_{2*[l]}), \langle \mathbf{P}(\Delta), \mathbf{P}(\Delta)\rangle_{2*[l]} +$$

$$\|\langle \mathbf{A}^{\otimes l}, \langle \mathbf{P}(\hat{\mathbf{w}}), \mathbf{P}(\Delta)\rangle_{2*[l]} + \langle \mathbf{P}(\Delta), \mathbf{P}(\hat{\mathbf{w}})\rangle_{2*[l]}\|_F^2 \geq 0 \quad \forall \Delta \in \mathbb{R}^{nr\circ l} \tag{23b}$$

*with* (23b) *being a necessary and sufficient condition for $\hat{\mathbf{w}}$ to be a FOP and $\nabla f_w^l(\mathbf{M})$ is defined as*

$$\nabla f_w^l(\mathbf{M}) = \langle (\mathbf{A}^{\otimes l})^* \mathbf{A}^{\otimes l}, \mathbf{M}\rangle - [\langle \mathbf{A}^*\mathbf{A}, M^*\rangle + \langle \mathbf{A}^*, \tilde{w}\rangle]^{\otimes l} \tag{24}$$

The proof to this lemma is highly technical and can be obtained by slightly changing the proof to Lemma 7 in Ma et al. (2024) by changing $b = \mathcal{A}(M^*)$ to $\tilde{b} = \mathcal{A}(M^*) + \tilde{w}$ defined above.

Here we present the full theorem of Theorem 3.2 regarding implicit bias in (12):

**Theorem C.9.** *Consider a finite-horizon gradient descent trajectory $\{\mathbf{w}_t\}_{t\in[T]}$ of* (12) *with $\mathbf{w}_{t+1} = \mathbf{w}_t - \eta\nabla h^l(\mathbf{w}_t)$ starting from the initialization $\mathbf{w}_0 = \xi x_0^{\otimes l}$ with $\xi \in \mathbb{R}$ denoting the scale of the initialization, $\eta$ representing the step-size and $x_0 \in \mathbb{R}^{nr}$ being an arbitrary vector with $\|x_0\|_2^2 = 1$. Then there exists $t(\kappa, l) \geq 1$ and $\kappa < 1$ such that*

$$\frac{\lambda_2^v(\mathbf{w}_t)}{\lambda_1^v(\mathbf{w}_t)} \leq \kappa, \qquad \forall t \in [t(\kappa, l), T] \tag{25}$$

*if the initialization scale $\xi$ is sufficiently small, where $t(\kappa, l)$ is expressed as*

$$t(\kappa, l) = \left\lceil \ln\left(\frac{\|x_0\|_2^l}{\kappa |v_1^\top x_0|^l}\right) \ln\left(\frac{1 + \eta\sigma_1^l(U)}{1 + \eta\sigma_2^l(U)}\right)^{-1} \right\rceil \tag{26}$$

*where $\sigma_1(U)$ and $\sigma_2(U)$ denote the first and second singular values of $U$ and $v_1, v_2$ are the associated singular vectors, with*

$$U = \langle \mathbf{A}_r, \tilde{b}\rangle_1 \in \mathbb{R}^{nr \times nr}, \quad \mathbf{A}_r \coloneqq I_r \oslash_{2,3} \mathbf{A} \tag{27}$$

Next, we present a technical extension of Theorem 4.1 and Theorem C.9, showing how gradient descent initialized with small scale can help ensure that second-order points of lifted version of (7) remain very close to $M^*$ along the optimization trajectory

**Theorem C.10.** *Consider a generic matrix completion problem under the same premise as given in Theorem 4.1. Assume that the symmetric tensor $\hat{\mathbf{w}} \in \mathbb{R}^{nr\circ l}$ is a second-order point (local minima) of* (12) *that is $\kappa$-rank-1 with $\kappa \leq \mathcal{O}(1/\|M^*\|_F^2)$. This can be achieved by initializing the vanilla gradient algorithm at $\mathbf{w}_0 = \xi x_0^{\otimes l}$ with a sufficiently small $\xi > 0 \in \mathbb{R}$. Then after iterations $t(\kappa, l)$ given in* (26)*, Theorem C.9 ensures that all tensors along the trajectory will become $\kappa$-rank-1.*

*If $\hat{\mathbf{w}}$'s major spectral decomposition is given as $\hat{\mathbf{w}} = \lambda_S \hat{x}^{\otimes l} + \hat{\mathbf{w}}^\dagger$ with $\hat{x} \in \mathbb{R}^{nr}$ being a FOP of* (7) *(ensured by Proposition 2 in Ma et al. (2024)), we know that*

$$\|M^* - \hat{X}\hat{X}^\top\|_F < \frac{1}{\epsilon}\lambda_r(\hat{X})\sqrt{\text{tr}(M^*)} + \mathcal{O}(\sqrt{r}\kappa^{1/2l}) + e_1 \tag{28}$$

*holds with probability at least $q$, under the condition that $l$ is odd and meets the requirement:*

$$l > \frac{1}{1 - \log_2(2\beta)}, \quad \beta \coloneqq \frac{\text{tr}(M^*)\lambda_r(\hat{X}\hat{X}^\top)}{\epsilon^2\left(\|M^* - \hat{X}\hat{X}^\top\|_F^2 - \mathcal{O}(r\kappa^{1/l}) - e_2\right)}. \tag{29}$$

*where $e_1$, $e_2$, and $q$ are identical to those given in Theorem 4.1 depending on different MC instances.*

The proof of this Theorem is omitted because it directly follows from Theorem 4.1, Theorem C.9, and Theorem 2 in Ma et al. (2024).

# D   MISSING PROOFS

**Lemma D.1** (RIP/RSC of $\epsilon$-MC). *Let $\mathcal{A}_\Omega : \mathbb{R}^{n \times n} \to \mathbb{R}^m$ be the sensing operator associated with a matrix completion problem, and let $\mathcal{A}_{\Omega,\epsilon}$ be its $\epsilon$–perturbed version defined as in (5) for some $\epsilon \in (0, 1]$. Then the corresponding $\epsilon$-MC objective*

$$f_\epsilon(M) \;=\; \|\mathcal{A}_{\Omega,\epsilon}(M)\|_2^2$$

*satisfies the $(1, n)$–RSS property and the $(\epsilon^2, n)$–RSC property, i.e.,*

$$\epsilon^2 \, \|M\|_F^2 \;\leq\; \|\mathcal{A}_{\Omega,\epsilon}(M)\|_2^2 \;\leq\; \|M\|_F^2 \qquad \text{for all } M \in \mathbb{R}^{n \times n}.$$

*In addition, suppose each entry of the original matrix completion problem is independently observed with probability $p \in (0, 1]$, and let $P_\Omega$ denote the sampling projector. Then*

$$\mathbb{E}\big[\|\mathcal{A}_{\Omega,\epsilon}(M)\|_2^2\big] \;=\; \big(p + \epsilon^2(1 - p)\big)\, \|M\|_F^2,$$

*so that the RSC constant improves in expectation from $\epsilon^2$ to $p + \epsilon^2(1 - p)$.*

*Moreover, with probability at least $1 - \delta$,*

$$\|\mathcal{A}_{\Omega,\epsilon}(M)\|_2^2 \geq \left( \epsilon^2 + (1 - \epsilon^2) \left[ p - \alpha_{\mathrm{sp}}(M)^2 \frac{(1 + 2p)\log(1/\delta)}{n^2} \right] \right) \|M\|_F^2$$

*where the spikiness of $M$ is defined as*

$$\alpha_{\mathrm{sp}}(M) = \frac{n\, \|M\|_{\max}}{\|M\|_F}, \quad \|M\|_{\max} := \max_{i,j} |M_{ij}|.$$

*Proof to Lemma D.1.* By construction of $\mathcal{A}_{\Omega,\epsilon}$ (see (5)), we have that

$$\|\mathcal{A}_{\Omega,\epsilon}(M)\|_2^2 = \sum_{(i,j)\in\Omega} M_{ij}^2 + \epsilon^2 \sum_{(i,j)\notin\Omega} M_{ij}^2$$

$$= \epsilon^2 \|M\|_F^2 + (1 - \epsilon^2)\, \|P_\Omega(M)\|_F^2,$$

where $P_\Omega$ is the sampling projector that keeps entries in $\Omega$ and zeros out the rest.

**Global RSS/RSC.**   For any $M$,

$$0 \;\leq\; \|P_\Omega(M)\|_F^2 \;\leq\; \|M\|_F^2.$$

Plugging these bounds into the decomposition above, we obtain

$$\epsilon^2 \|M\|_F^2 \;\leq\; \epsilon^2 \|M\|_F^2 + (1 - \epsilon^2)\|P_\Omega(M)\|_F^2 \;\leq\; \epsilon^2 \|M\|_F^2 + (1 - \epsilon^2)\|M\|_F^2 = \|M\|_F^2.$$

This yields the global $(\epsilon^2, n)$–RSC and $(1, n)$–RSS inequalities.

**Expected RSC constant under Bernoulli sampling.**   Now assume each entry is observed independently with probability $p \in (0, 1]$, so $S_{ij} \sim \mathrm{Bernoulli}(p)$ and

$$P_\Omega(M)_{ij} = S_{ij} M_{ij}.$$

Then

$$\mathbb{E}\|P_\Omega(M)\|_F^2 = \mathbb{E} \sum_{i,j} S_{ij} M_{ij}^2 = p \sum_{i,j} M_{ij}^2 = p\|M\|_F^2.$$

Taking expectations in the decomposition

$$\|\mathcal{A}_{\Omega,\epsilon}(M)\|_2^2 = \epsilon^2 \|M\|_F^2 + (1 - \epsilon^2)\|P_\Omega(M)\|_F^2$$

gives

$$\mathbb{E}\big[\|\mathcal{A}_{\Omega,\epsilon}(M)\|_2^2\big] = \epsilon^2 \|M\|_F^2 + (1 - \epsilon^2)\, p\|M\|_F^2 = \big(p + \epsilon^2(1 - p)\big)\, \|M\|_F^2,$$

which is the claimed improvement of the RSC constant in expectation.

**High-probability RSC via spikiness.** To obtain a meaningful deviation guarantee from the afore-mentioned mean, we control $\|P_\Omega(M)\|_F^2$ via a scalar Bernstein inequality and the spikiness of $M$.

Write

$$\|P_\Omega(M)\|_F^2 = \sum_{i,j} S_{ij} M_{ij}^2.$$

Define

$$X_{ij} := (S_{ij} - p) M_{ij}^2, \qquad X := \sum_{i,j} X_{ij} = \|P_\Omega(M)\|_F^2 - p\|M\|_F^2.$$

Each $X_{ij}$ is mean-zero and independent. Let $B := \|M\|_{\max}^2$. Then

$$|X_{ij}| \le B, \qquad \mathrm{Var}(X_{ij}) = p(1-p)M_{ij}^4 \le pM_{ij}^4.$$

Hence the variance proxy

$$\mathrm{Var}(X) = \sum_{i,j} \mathrm{Var}(X_{ij}) \ \le \ p\sum_{i,j} M_{ij}^4 \ \le \ p\|M\|_{\max}^2 \|M\|_F^2 = pB\|M\|_F^2.$$

Using a standard one-sided Bernstein-type inequality for a random variable $X$ satisfying a Bernstein condition with parameter $b$,

$$\mathbb{P}\Big( X \ge \mathbb{E}X - \big(b\log(1/\delta) + 2\,\mathrm{Var}(X)\,\log(1/\delta)\big)\Big) \ \ge \ 1 - \delta,$$

and applying it to $X = \|P_\Omega(M)\|_F^2 - p\|M\|_F^2$ with $b = B$ and $\mathrm{Var}(X) \le pB\|M\|_F^2$, we obtain, with probability at least $1 - \delta$,

$$\|P_\Omega(M)\|_F^2 \ \ge \ p\|M\|_F^2 - B\log(1/\delta) - 2pB\|M\|_F^2 \log(1/\delta).$$

Dividing by $\|M\|_F^2$ yields

$$\frac{\|P_\Omega(M)\|_F^2}{\|M\|_F^2} \ \ge \ p - \frac{B}{\|M\|_F^2}\log(1/\delta) - 2p\frac{B}{\|M\|_F^2}\log(1/\delta).$$

By the spikiness definition,

$$\alpha_{\mathrm{sp}}(M)^2 = \frac{N\|M\|_{\max}^2}{\|M\|_F^2} = \frac{n^2 B}{\|M\|_F^2} \quad\Longrightarrow\quad \frac{B}{\|M\|_F^2} = \frac{\alpha_{\mathrm{sp}}(M)^2}{n^2},$$

so we obtain, with probability at least $1 - \delta$,

$$\frac{\|P_\Omega(M)\|_F^2}{\|M\|_F^2} \ \ge \ p - \alpha_{\mathrm{sp}}(M)^2 \frac{(1+2p)\log(1/\delta)}{n^2}.$$

Finally, substituting this bound back into the decomposition of $\|\mathcal{A}_{\Omega,\epsilon}(M)\|_2^2$ gives

$$\frac{\|\mathcal{A}_{\Omega,\epsilon}(M)\|_2^2}{\|M\|_F^2} = \epsilon^2 + (1-\epsilon^2)\frac{\|P_\Omega(M)\|_F^2}{\|M\|_F^2}$$

$$\ge \epsilon^2 + (1-\epsilon^2)\left[p - \alpha_{\mathrm{sp}}(M)^2\frac{(1+2p)\log(1/\delta)}{n^2}\right] \ =: \ \alpha_s,$$

which shows that, with probability at least $1 - \delta$, the $\epsilon$-MC operator enjoys an $(\alpha_s, n)$–RSC property with $\alpha_s$ as stated in the lemma. $\qquad\square$

*Proof to Theorem 2.2.* To begin with, we reiterate our elementary results which follows from the definition of $M^\dagger$ and that of (7):

$$0 \ge f_{w_\epsilon}(M^\dagger) - f_{w_\epsilon}(M^*)$$

$$= \frac{1}{2}\|\mathcal{A}_{\Omega,\epsilon}(M^\dagger - M^*)\|_{\tilde{S},2}^2 + \frac{1}{2}\|\mathcal{A}_{\Omega,\epsilon}(M^\dagger - M^*) - w_\epsilon\|_{S,2}^2 - \frac{1}{2}\|w_\epsilon\|_{S,2}^2$$

$$= \frac{1}{2}\|\mathcal{A}_{\Omega,\epsilon}(M^\dagger - M^*)\|_{\tilde{S},2}^2 + \frac{1}{2}\|\mathcal{A}_{\Omega,\epsilon}(M^\dagger)\|_{S,2}^2 - \frac{1}{2}\epsilon^2\|M^*\|_{S,2}^2$$

$$\ge \frac{1}{2}\|\mathcal{A}_{\Omega,\epsilon}(M^\dagger - M^*)\|_{\tilde{S},2}^2 - \frac{1}{2}\epsilon^2\|M^*\|_{S,2}^2$$

This follows from the simple observation that

$$(w_\epsilon)_i = -\epsilon \operatorname{vec}(M^*)_i \quad \forall \ i \in [n^2]$$

Then moving $\frac{1}{2}\epsilon^2 \|M^*\|_{S,2}^2$ to the left hand side, and adding $\frac{1}{2}\|M^\dagger - M^*\|_{S,2}^2$ to both sides gives

$$\|M^\dagger - M^*\|_F^2 \leq \|M^\dagger - M^*\|_S^2 + \epsilon^2 \|M^*\|_{S,2}^2 \tag{30}$$

If we define a new vector $d \in \mathbb{R}^{n^2}$ in which

$$d := \operatorname{vec}(M^\dagger - M^*) \odot \operatorname{vec}(M^\dagger - M^*) + \epsilon^2 \operatorname{vec}(M^*) \odot \operatorname{vec}(M^*)$$

then we know that

$$d_i = \left(\operatorname{vec}(M^\dagger)_i - \operatorname{vec}(M^*)_i\right)^2 + \epsilon^2 \operatorname{vec}(M^*)_i^2 \geq 0 \quad \forall \ i \in [n^2]$$

So if we further define a series of random variables $\{r_1, r_2, \ldots, r_{n^2}\}$ with

$$r_i = \begin{cases} 0 & \text{with probability } p \\ d_i & \text{with probability } 1-p \end{cases} \tag{31}$$

Then we know that

$$\|M^\dagger - M^*\|_S^2 + \epsilon^2 \|M^*\|_{S,2}^2 = \sum_{i=1}^{n^2} r_i := R \tag{32}$$

because for any matrix $M \in \mathbb{R}^{n_1 \times n_2}$, we have

$$\|M\|_S^2 = \sum_i^{n_1 n_2} m_i^2, \quad m_i = \begin{cases} 0 & \text{with probability } p \\ \operatorname{vec}(M)_i & \text{with probability } 1-p \end{cases}$$

Then we simply acknowledge that $0 \leq r_i \leq d_i$ almost surely, which sets up the premise to use Hoeffding's inequality (Hoeffding, 1994). This concentration inequality gives that

$$\mathbb{P}\left(R \leq \mathbb{E}[R] + t\right) \geq 1 - \exp\left(\frac{-2t^2}{\sum_i^{n^2}(d_i - 0)^2}\right) = 1 - \exp\left(\frac{-2t^2}{\|d\|_2^2}\right) \tag{33}$$

First of all, we could easily derive that

$$\begin{aligned} \mathbb{E}[R] &= \sum_i^{n^2}(1-p)\left[\left(\operatorname{vec}(M^\dagger)_i - \operatorname{vec}(M^*)_i\right)^2 + \epsilon^2 \operatorname{vec}(M^*)_i^2\right] \\ &= (1-p)\left(\|M^\dagger - M^*\|_F^2 + \epsilon^2 \|M^*\|_F^2\right) \end{aligned} \tag{34}$$

Therefore combining (30), (33) and (34) we have

$$\mathbb{P}\left(\|M^\dagger - M^*\|_F^2 \leq (1-p)\left(\|M^\dagger - M^*\|_F^2 + \epsilon^2 \|M^*\|_F^2\right) + t\right) \geq 1 - \exp\left(\frac{-2t^2}{\|d\|_2^2}\right) \tag{35}$$

Then we can choose

$$t = \chi\left(\|M^\dagger - M^*\|_F^2 + \epsilon^2 \|M^*\|_F^2\right) = \chi\|d\|_1$$

for some constant $\chi \leq p$. This will then transform (35) into

$$\mathbb{P}\left(\|M^\dagger - M^*\|_F^2 \leq (1-p+\chi)\left(\|M^\dagger - M^*\|_F^2 + \epsilon^2 \|M^*\|_F^2\right)\right) \geq 1 - \exp\left(\frac{-2t^2}{\|d\|_2^2}\right)$$

$$\implies \mathbb{P}\left((p-\chi)\|M^\dagger - M^*\|_F^2 \leq (1-p+\chi)\epsilon^2 \|M^*\|_F^2\right) \geq 1 - \exp\left(\frac{-2\chi^2\|d\|_1^2}{\|d\|_2^2}\right) \tag{36}$$

$$\implies \mathbb{P}\left(\|M^\dagger - M^*\|_F^2 \leq \frac{1-p+\chi}{p-\chi}\epsilon^2 \|M^*\|_F^2\right) \geq 1 - \exp\left(\frac{-2\chi^2\|d\|_1^2}{\|d\|_2^2}\right)$$

which proves our desired result directly. $\qquad\square$

*Proof of Theorem 3.1.* First of all, we hope to decompose the Hessian of (1) at a second order point $\hat{X} \in \mathbb{R}^{n \times r}$. Classic matrix sensing literatures like Ha et al. (2020); Zhang et al. (2021); Li et al. (2019) give that the second-order critical condition of (1) are given as

$$\nabla f(\hat{X}\hat{X}^\top)\hat{X} = 0, \tag{37}$$

$$2\langle \nabla f(\hat{X}\hat{X}^\top), UU^\top \rangle + [\nabla^2 f(\hat{X}\hat{X}^\top)](\hat{X}U^\top + U\hat{X}^\top, \hat{X}U^\top + U\hat{X}^\top) \geq 0 \quad \forall U \in \mathbb{R}^{n \times r} \tag{38}$$

with (37) being the first order critical condition. Moreover, since the sensing matrices $\{A_i\}_{i \in [m]}$ can be assumed be to symmetric without loss of generality (Zhang et al., 2021), we have that

$$[\nabla^2 f(\hat{X}\hat{X}^\top)](\hat{X}U^\top + U\hat{X}^\top, \hat{X}U^\top + U\hat{X}^\top) = 4[\nabla^2 f(\hat{X}\hat{X}^\top)](\hat{X}U^\top, \hat{X}U^\top).$$

We then could decompose LHS of (38) as $2C_1 + 4C_2$ where

$$C_1 := \langle \nabla f(\hat{X}\hat{X}^\top), UU^\top \rangle, \quad C_2 := [\nabla^2 f(\hat{X}\hat{X}^\top)](\hat{X}U^\top, \hat{X}U^\top)$$

Given the assumption that (1) obeys some RSS condition, it is possible to upper-bound $C_2$ by observing

$$[\nabla^2 f(\hat{X}\hat{X}^\top)](\hat{X}U^\top + U\hat{X}^\top, \hat{X}U^\top + U\hat{X}^\top) \leq L_s \|\hat{X}U^\top + U\hat{X}^\top\|_F^2$$

Therefore, if want to somehow create an negative escape direction for $\hat{X}$, it is important that we find a $U$ such that $C_1$ is negative and large in magnitude, and then amplify this term via tensor parametrization. To do so, we first do a more in-depth analysis of $\nabla f(\hat{X}\hat{X}^\top)$. As mentioned above, since $\nabla f(\cdot)$ can be assumed to be symmetric, one can select $u \in \mathbb{R}^n$ such that $u^\top \nabla f(\hat{x}\hat{x}^\top)u = \lambda_{\min}(\nabla f(\hat{x}\hat{x}^\top))$. Then via the definition of RSC we have

$$f(M^*) \geq f(\hat{X}\hat{X}^\top) + \langle \nabla f(\hat{X}\hat{X}^\top), M^* - \hat{X}\hat{X}^\top \rangle + \frac{\alpha_s}{2}\|\hat{X}\hat{X}^\top - M^*\|_F^2. \tag{39}$$

With $\hat{X}$ being a first-order point, according to (37)

$$\nabla f(\hat{X}\hat{X}^\top)\hat{X} = 0 \implies \langle \nabla f(\hat{X}\hat{X}^\top), \hat{X}\hat{X}^\top \rangle = 0$$

Therefore, if in (1) our $b$ is corrupted as $\mathcal{A}(M^*) + \tilde{w}$, then plugging it back into (39) gives

$$\begin{aligned}
\langle \nabla f(\hat{X}\hat{X}^\top), M^* \rangle &\leq -\frac{\alpha_s}{2}\|\hat{x}\hat{x}^\top - M^*\|_F^2 + f(M^*) - f(XX^\top) \\
&\leq -\frac{\alpha_s}{2}\|\hat{x}\hat{x}^\top - M^*\|_F^2 + f(M^*) \\
&= -\frac{\alpha_s}{2}\|\hat{x}\hat{x}^\top - M^*\|_F^2 + \frac{\|\tilde{w}\|_2^2}{2}
\end{aligned} \tag{40}$$

where the second inequality follows from the fact that $f(\cdot) \geq 0$ in its entire domain and the last inequality follows from $f(M^*) = 1/2\|\mathcal{A}(M^*) - \mathcal{A}(M^*) - \tilde{w}\|_2^2 = \|\tilde{w}\|_2^2/2$. Furthermore, since both $\nabla f(\hat{X}\hat{X}^\top)$ and $M^*$ are assumed to be positive semidefinite,

$$\langle \nabla f(\hat{X}\hat{X}^\top), M^* \rangle \geq \lambda_{\min}(\nabla f(\hat{X}\hat{X}^\top)) \operatorname{tr}(M^*)$$

which implies that

$$\lambda_{\min}(\nabla f(\hat{X}\hat{X}^\top)) \leq \frac{-\alpha_s\|\hat{X}\hat{X}^\top - M^*\|_F^2 + \|\tilde{w}\|_2^2}{2\operatorname{tr}(M^*)} \tag{41}$$

Furthermore, (13) gives us

$$\|\hat{X}\hat{X}^\top - M^*\|_F^2 \geq \|\tilde{w}\|_2^2/\alpha_s$$

since $\frac{L_s}{\alpha_s}\lambda_r(\hat{X}\hat{X}^\top)\operatorname{tr}(M^*) \geq 0$ by definition. This means that

$$\lambda_{\min}(\nabla f(\hat{X}\hat{X}^\top)) \leq \frac{-\alpha_s\|\hat{X}\hat{X}^\top - M^*\|_F^2 + \|\tilde{w}\|_2^2}{2\operatorname{tr}(M^*)} \leq 0 \tag{42}$$

Thus, with this result equipped, we can further find a $U$ that makes $C_1$ small. In the most convenient manner, we first consider the eigenvector $u \in \mathbb{R}^n$ of $\nabla f(\hat{X}\hat{X}^\top)$ associated with $\lambda_{\min}(\nabla f(\hat{X}\hat{X}^\top))$. Additionally we consider $q \in \mathbb{R}^r$ to be the $r$-th singular value of $\hat{X}$, with

$$\|\hat{X}q\|_2 = \sigma_r(\hat{X}), \qquad \|q\|_2 = 1$$

Then choosing $U \in \mathbb{R}^{n \times r} = u q^\top$ leads to

$$C_1 = \langle \nabla f(\hat{X}\hat{X}^\top), U U^\top \rangle = \langle \nabla f(\hat{X}\hat{X}^\top), u u^\top \rangle = -G$$

where $G := -\lambda_{\min}(\nabla f(\hat{X}\hat{X}^\top)) \geq 0$. By recalling $\hat{X}^\top u = 0$ according to the first-order condition (37), we can further bound $C_2$ with this chocie of $U$ as

$$\begin{aligned}
L_s \|\hat{X}U^\top + U\hat{X}^\top\|_F^2 &= L_s \|u(\hat{X}q)^\top + (\hat{X}q)u^\top\|_F^2 \\
&= 2L_s \|\hat{X}q\|_F^2 + 2L_s (q^\top(\hat{X}^\top u))^2 \\
&= 2L_s \lambda_r(\hat{X}\hat{X}^\top),
\end{aligned}$$

leading to

$$C_2 \leq \frac{1}{2} L_s \lambda_r(\hat{X}\hat{X}^\top)$$

Now, if we choose $\Delta = \text{vec}(U)^{\otimes l}$ for the aforementioned $U \in \mathbb{R}^{n \times r}$, the LHS of (23b) can be expressed as:

$$2(\langle \mathbf{A}, \hat{X}\hat{X}^\top \rangle_{2,3}^\top \langle \mathbf{A}, u u^\top \rangle_{2,3})^l - 2\left((\langle \mathbf{A}, M^* \rangle_{2,3} + \tilde{w})^\top \langle \mathbf{A}, u u^\top \rangle_{2,3}\right)^l + 4(\|\langle \mathbf{A}, \hat{X}U^\top \rangle_{2,3}\|_2^2)^l$$

$$\leq 2(\lambda_{\min}(\nabla f(\hat{X}\hat{X}^\top)))^l + 4C_2^l$$

$$= 2C_1^l + 4C_2^l \tag{43}$$

where the inequality follows from:

$$a^n - b^n \leq (a-b)^n, \quad \forall b \geq a \geq 0$$

Here, since $a - b = C_1 \leq 0$, the above inequality can be used. As a result,

$$\text{LHS of (23b)} \leq \underbrace{-2G^l}_{\text{Part 1}} + \underbrace{\frac{2}{2^{l-1}} L_s^l \lambda_r(\hat{X}\hat{X}^\top)^l}_{\text{Part 2}}$$

We know since $G \geq 0$, Part 1 is always negative assuming $l$ is odd, and Part 2 is always positive. Therefore, it suffices to find an order $l$ such that

$$G^l > (1/2^{l-1}) L_s^l \lambda_r(\hat{X}\hat{X}^\top)^l \tag{44}$$

Conveniently, (42) says that

$$G \geq \frac{\alpha_s \|M^* - \hat{X}\hat{X}^\top\|_F^2 - \|\tilde{w}\|_2^2}{2\,\text{tr}(M^*)}, \tag{45}$$

which can be used to derive sufficient condition for (44). Therefore, if

$$\left(\frac{\alpha_s \|M^* - \hat{X}\hat{X}^\top\|_F^2 - \|\tilde{w}\|_2^2}{2\,\text{tr}(M^*)}\right)^l > (1/2^{l-1}) L_s^l \lambda_r(\hat{X}\hat{X}^\top)^l,$$

we can conclude that (44) holds, which implies that the LHS of (23b) is negative, directly proving that $\hat{X}^{\otimes l}$ is not an SOP anymore. Elementary manipulations of the above equation give that a sufficient condition is

$$\|M^* - \hat{X}\hat{X}^\top\|_F^2 - \|\tilde{w}\|_2^2/\alpha_s > 2^{1/l} \frac{L_s}{\alpha_s} \lambda_r(\hat{X}\hat{X}^\top) \,\text{tr}(M^*) \tag{46}$$

We now consider (13), which means that

$$\lambda_r(\hat{X}\hat{X}^\top) \leq \frac{\alpha_s \|M^* - \hat{X}\hat{X}^\top\|_F^2 - \|\tilde{w}\|_2^2}{L_s \,\text{tr}(M^*)} \tag{47}$$

Subsequently, define a constant $\gamma$ such that

$$L_s \lambda_r(\hat{X}\hat{X}^\top) = \gamma \left[\frac{\alpha_s \|M^* - \hat{X}\hat{X}^\top\|_F^2 - \|\tilde{w}\|_2^2}{2\,\text{tr}(M^*)}\right]$$

Then, (45) and (47) together imply that $1 \leq \gamma < 2$. Using this simplified notation, our sufficient condition (46) becomes

$$1 > \frac{\gamma}{2^{(l-1)/l}} \tag{48}$$

Given $1 \leq \gamma < 2$, there always exists a large enough $l$ such that (48) holds, which proves that LHS of (23b) is negative, proving that $\text{vec}(\hat{X})^{\otimes l}$ is a strict saddle, concluding the proof.

To derive a sufficient $l$, we simply acknowledge

$$\gamma = \frac{2L_s \text{tr}(M^*)\lambda_r(\hat{X}\hat{X}^\top)}{\alpha_s \|M^* - \hat{X}\hat{X}^\top\|_F^2 - \|\tilde{w}\|_2^2} := 2\beta$$

and that $\beta \leq 1$ due to assumption (13). Therefore, for (48) to hold true, it is enough to have

$$2^{(l-1)/l} > 2\beta \implies \frac{l-1}{l} > \log_2(2\beta) \implies l > \frac{1}{1 - \log_2(2\beta)}$$

$\square$

*Proof of Theorem C.9.* First of all, we hope to decompose the GD trajectory of (12) $\{\mathbf{w}_t\}_{t=0}^T$ as follows:

$$\mathbf{w}_{t+1} = \langle \mathbf{Z}_t, \mathbf{w}_0 \rangle - \mathbf{E}_t := \tilde{\mathbf{w}}_t - \mathbf{E}_t \tag{49}$$

where

$$\mathbf{Z}_t := (\mathcal{I} + \eta\langle \mathbf{A}_r^{\otimes l}, \tilde{b}^{\otimes l}\rangle)^t, \quad \mathbf{A}_r = I_r \oslash_{2,3} A$$

$$\mathbf{E}_t := \sum_{i=1}^t (\mathcal{I} + \eta\langle \mathbf{A}_r^{\otimes l}, \tilde{b}^{\otimes l}\rangle)^{t-i}\hat{\mathbf{E}}_i$$

$$\hat{\mathbf{E}}_i := \eta\langle\langle (\mathbf{A}_r^l)^*\mathbf{A}^l, \langle \mathbf{P}(\mathbf{w}_{i-1}), \mathbf{P}(\mathbf{w}_{i-1})\rangle_{2*[l]}\rangle, \mathbf{w}_{i-1}\rangle_{2*[l]}$$

$$(\mathbf{A}_r^l)^*\mathbf{A}^l := \langle (\mathbf{A}_r)^{\otimes l}, \mathbf{A}^{\otimes l}\rangle_{3,6,\dots,3l} \in \mathbb{R}^{[nr \times nr \times n \times n]\circ l}$$

This can be proved via induction where

$$\mathbf{w}_1 = \left(\mathcal{I} + \eta\langle \mathbf{A}_r^{\otimes l}, \tilde{b}^{\otimes l} - (\mathbf{A}^{\otimes l})^*\langle \mathbf{P}(\mathbf{w}_0), \mathbf{P}(\mathbf{w}_0)\rangle\rangle\right)\mathbf{w}_0$$

$$= (\mathcal{I} + \eta\langle \mathbf{A}_r^{\otimes l}, \tilde{b}^{\otimes l}\rangle)\mathbf{w}_0 - \eta\langle (\mathbf{A}_r^l)^*\mathbf{A}^l, \langle \mathbf{P}(\mathbf{w}_0), \mathbf{P}(\mathbf{w}_0)\rangle\rangle\mathbf{w}_0$$

$$= \langle \mathbf{Z}_1, \mathbf{w}_0\rangle - \mathbf{E}_1$$

This serves as our base case, and the induction step can be proven as

$$\mathbf{w}_{t+1} = \left(\mathcal{I} + \eta\langle \mathbf{A}_r^{\otimes l}, \tilde{b}^{\otimes l} - (\mathbf{A}^{\otimes l})^*\langle \mathbf{P}(\mathbf{w}_t), \mathbf{P}(\mathbf{w}_t)\rangle\rangle\right)\mathbf{w}_t$$

$$= (\mathcal{I} + \eta\langle \mathbf{A}_r^{\otimes l}, \tilde{b}^{\otimes l}\rangle)\mathbf{w}_t - \eta\langle (\mathbf{A}_r^l)^*\mathbf{A}^l, \langle \mathbf{P}(\mathbf{w}_t), \mathbf{P}(\mathbf{w}_t)\rangle\rangle\mathbf{w}_t$$

$$= (\mathcal{I} + \eta\langle \mathbf{A}_r^{\otimes l}, \tilde{b}^{\otimes l}\rangle)\mathbf{w}_t - \hat{\mathbf{E}}_{t+1}$$

$$= (\mathcal{I} + \eta\langle \mathbf{A}_r^{\otimes l}, \tilde{b}^{\otimes l}\rangle)\left(\tilde{\mathbf{w}}_t - \sum_{i=1}^t (\mathcal{I} + \eta\langle \mathbf{A}_r^{\otimes l}, \tilde{b}^{\otimes l}\rangle)^{t-i}\hat{\mathbf{E}}_i\right) - \hat{\mathbf{E}}_{t+1}$$

$$= \tilde{\mathbf{w}}_{t+1} - \sum_{i=1}^t (\mathcal{I} + \eta\langle \mathbf{A}_r^{\otimes l}, \tilde{b}^{\otimes l}\rangle)^{t+1-i}\hat{\mathbf{E}}_i - \hat{\mathbf{E}}_{t+1}$$

$$= \tilde{\mathbf{w}}_{t+1} - \sum_{i=1}^{t+1} (\mathcal{I} + \eta\langle \mathbf{A}_r^{\otimes l}, \tilde{b}^{\otimes l}\rangle)^{t+1-i}\hat{\mathbf{E}}_i$$

$$= \tilde{\mathbf{w}}_{t+1} - \mathbf{E}_t$$

Therefore, we can then use a version of Lemma 13 and Lemma 2 in Ma et al. (2024) with $U = \langle \mathbf{A}_r^*\mathbf{A}, M^*\rangle$ replaced with $U = \langle \mathbf{A}_r, \tilde{b}\rangle_1$ to prove this theorem, following the steps in proof to Theorem 1 in Ma et al. (2024). $\square$

*Proof to Theorem 4.1.* For the general matrix completion case, as promised by Lemma 2.1, the theorem is proved by substituting $L_s = 1$, $\alpha_s = \epsilon^2$ into Theorem 4.1, and since this is a deterministic result, it happens with probability 1, meaning that for any second-order point $\hat{\mathbf{w}} = \hat{X}^{\otimes l}$ of (12), it satisfies that

$$
\begin{aligned}
\|M^* - \hat{X}\hat{X}^\top\|_F &< \sqrt{\frac{L_s}{\alpha_s}\lambda_r(\hat{X}\hat{X}^\top)\operatorname{tr}(M^*) + \frac{\epsilon^2\|M^*\|_{\bar{\Omega},F}^2}{\epsilon^2}} \\
&= \sqrt{\frac{L_s}{\alpha_s}\lambda_r(\hat{X}\hat{X}^\top)\operatorname{tr}(M^*) + \|M^*\|_{\bar{\Omega},F}^2} \\
&\leq \sqrt{\frac{L_s}{\alpha_s}\lambda_r(\hat{X}\hat{X}^\top)\operatorname{tr}(M^*)} + \|M^*\|_{\bar{\Omega},F} \\
&= \frac{1}{\epsilon}\lambda_r(\hat{X})\sqrt{\operatorname{tr}(M^*)} + \|M^*\|_{\bar{\Omega},F}
\end{aligned}
\tag{50}
$$

where $l$ has to obey equation (14) as stated in Theorem 3.1.

In the case of each entry of $M^*$ being observed independently with probability $p$, we first apply Theorem 4.1 with $\tilde{w} = 0$ to (7), meaning that we first assume that no noise exists in $b$. This is the case where we are actually trying to recover the global solution of (7), denoted as $M^\dagger$. This means that for any rank-1 critical point $\hat{\mathbf{w}} = \hat{X}^{\otimes l}$ of (12), it is a second-order point only if

$$
\|\hat{X}\hat{X}^\top - M^\dagger\|_F^2 < \frac{1}{\epsilon^2}\lambda_r(\hat{X}\hat{X}^\top)\operatorname{tr}(M^*)
\tag{51}
$$

holds, when $l$ is odd and satisfies

$$
l > \frac{1}{1 - \log_2(2\beta)}, \quad \beta := \frac{L_s\operatorname{tr}(M^*)\lambda_r(\hat{X}\hat{X}^\top)}{\epsilon^2\|M^* - \hat{X}\hat{X}^\top\|_F^2}.
\tag{52}
$$

The above statement holds deterministically. However, Theorem 2.2 also tells us that

$$
\|M^\dagger - M^*\|_F^2 \leq \frac{1 - p + \chi}{p - \chi}\epsilon^2\|M^*\|_F^2
\tag{}
$$

with high probability, so then by a triangle inequality we have that the conversion criterion above transforms to

$$
\begin{aligned}
\|\hat{X}\hat{X}^\top - M^*\|_F &\leq \|\hat{X}\hat{X}^\top - M^\dagger\|_F + \|M^\dagger - M^*\|_F \\
&< \frac{\lambda_r(\hat{X})\sqrt{\operatorname{tr}(M^*)}}{\epsilon} + \sqrt{\frac{1 - p + \chi}{p - \chi}}\epsilon\|M^*\|_F
\end{aligned}
\tag{53}
$$

with the same probability stated in Theorem 2.2, thereby concluding the proof. $\square$

# E SPATIAL DISTRIBUTION OF RECONSTRUCTION ERROR

Global metrics such as the Frobenius error $\|M^{\mathrm{sol}} - M^\star\|_F$ summarize the quality of reconstruction in a single number, but they do not reveal how errors are distributed across the matrix. In such cases, it is natural to ask whether failure manifests as a few "bad" rows or columns, or whether the algorithm fits the observed entries well but extrapolates incorrectly on a subset of unobserved entries.

To investigate this question, we analyze the entrywise squared error

$$
E_{ij}^2 = (M_{ij}^{\mathrm{sol}} - M_{ij}^\star)^2
$$

and its spatial distribution over the matrix. We consider the structured instance of problem (20) and use our method described in Theorem 4.1 to obtain reconstructed solutions $M^{\mathrm{sol}}$. We performed numerical studies on $E$, and have made some interesting observations. A striking feature of all unsuccessful runs is that the error on observed entries is essentially zero. The optimization procedure consistently recovers the observed data very accurately. The non-negligible reconstruction error is confined to unobserved entries, indicating that the algorithm converges to stationary points that are locally optimal with respect to the observed data, rather than diverging or producing a globally incoherent factorization.

| | s0 | s1 | s2 | s3 | s4 | s5 | s6 | s7 | s8 | s9 |
|---|---|---|---|---|---|---|---|---|---|---|
| $\text{MSE}_{\text{obs}}$ | 3.26e-8 | 4.89e-8 | 9.46e-9 | 9.79e-9 | 1.35e-8 | 1.67e-8 | 1.07e-8 | 9.40e-9 | 1.41e-8 | 1.50e-8 |
| $\text{MSE}_{\text{unobs}}$ | 2.67e0 | 4.95e-8 | 2.67e0 | 2.67e0 | 8.23e-8 | 8.24e-8 | 2.67e0 | 2.67e0 | 2.67e0 | 2.67e0 |

Table 2: Mean-squared error on observed and unobserved entries for the structured instance in problem (20) with $n = 8$, $\ell = 3$, $\epsilon = 10^{-4}$, learning rate 0.02, and initialization scale $10^{-4}$. Columns $s0$–$s9$ correspond to independent runs (`sol_0-sol_9`). In most runs, $\text{MSE}_{\text{obs}}$ is near machine precision while $\text{MSE}_{\text{unobs}}$ is either of order 1 (unsuccessful completions) or of order $10^{-8}$ (successful completions), highlighting that unsuccessful runs fit the observed entries but differ in their extrapolation to unobserved entries.

Table 2 showcases this phenomenon quantitatively. Across 10 independent runs, we report the mean-squared error on observed and unobserved entries, denoted $\text{MSE}_{\text{obs}}$ and $\text{MSE}_{\text{unobs}}$, respectively. In 10 out of 10 runs, $\text{MSE}_{\text{obs}}$ is on the order of $10^{-8}$ or even lower, while $\text{MSE}_{\text{unobs}}$ is solely responsible in determining whether the reconstruction was successful or not. This shows a gap of roughly eight orders of magnitude between the observed and unobserved regions. To take a closer look at the results, we selected 4 runs out of the 10 and plotted their entrywise reconstruction errors using heat-maps in Figure 4. We overlayed the observation mask $\Omega$ with white squares indicating the observed entries, while the background color encodes the squared reconstruction error.

From Figure 4 we observe that the error patterns on the unobserved entries differ substantially across the three unsuccessful runs. In each case, different subsets of the unobserved region carry the dominant error, and there is no single block or band that is systematically problematic across runs. This behavior is consistent with the presence of many spurious local minima in problem (20): different random initializations lead gradient descent to distinct low-rank completions that interpolate the observed entries but disagree on how to fill in the missing part. In contrast, the successful run exhibits uniformly small squared error on both observed and unobserved entries, corresponding to convergence to the global minimum. Taken together, these experiments suggest that unsuccessful recoveries are primarily due to convergence to benign local minima that fit the data but differ in their extrapolation to unobserved entries, rather than to systematic failure in any specific region of the matrix.

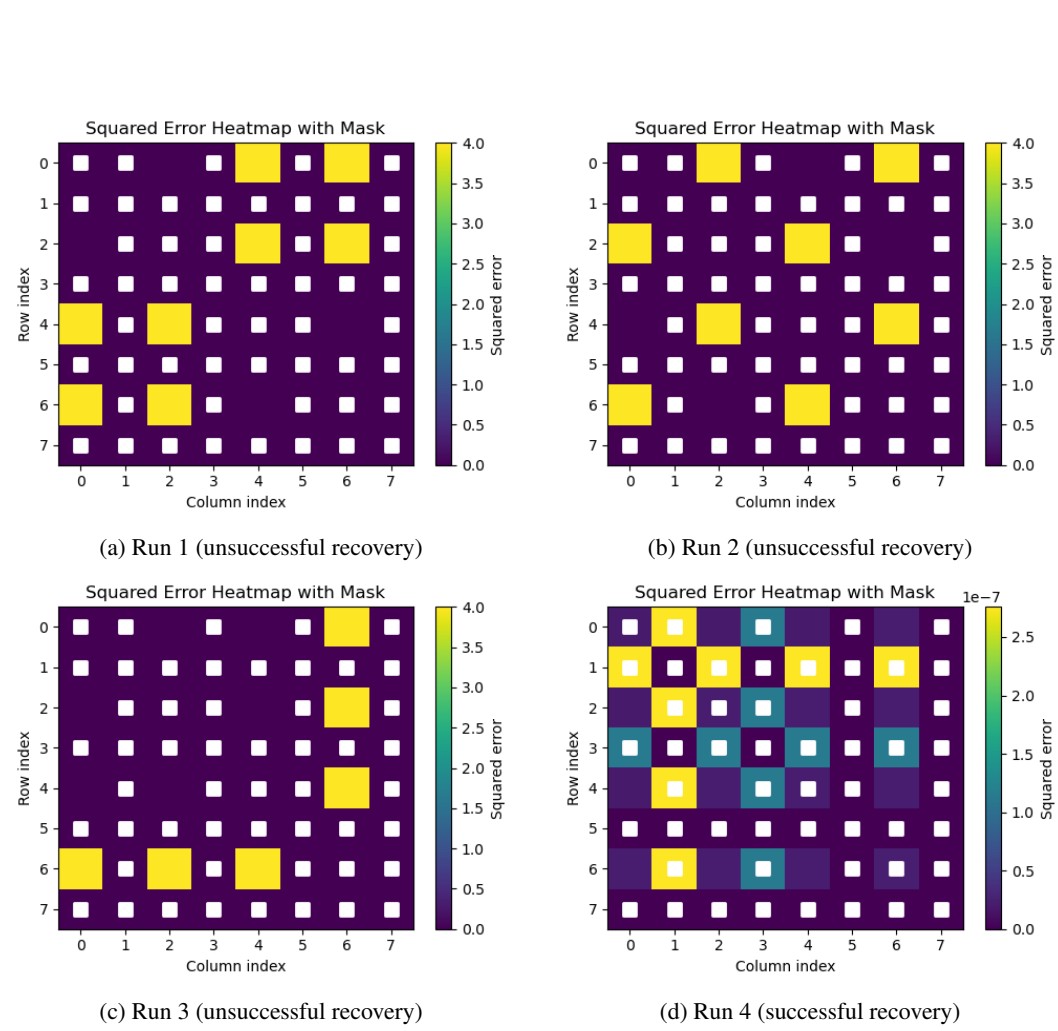

(a) Run 1 (unsuccessful recovery)

(b) Run 2 (unsuccessful recovery)

(c) Run 3 (unsuccessful recovery)

(d) Run 4 (successful recovery)

Figure 4: Spatial distribution of squared reconstruction error for the structured instance in problem (20) with $n = 8$, $\ell = 3$, $\epsilon = 10^{-4}$, learning rate 0.02, and initialization scale $10^{-4}$. The background colormap shows $(M_{ij}^{\mathrm{sol}} - M_{ij}^{\star})^2$, and white squares denote observed entries $(i, j) \in \Omega$. In the three runs with higher global error (a)–(c), the reconstruction is essentially exact on observed entries and the error is concentrated on unobserved entries, with different spatial patterns across runs. In the successful run (d), the error is small on both observed and unobserved entries.