# OpenReview forum: "Using Noise to Help Reach Global Minima: Turning Matrix Completion into Noisy Matrix Sensing"
_ICLR.cc/2026/Conference — Submitted to ICLR 2026_

### Official Review · Reviewer_oKEv · 2025-10-30

**Soundness:** 3
**Presentation:** 4
**Contribution:** 3
**Rating:** 8
**Confidence:** 3

**Summary:**

This paper studies a surrogate formulation of the classical matrix completion problem, which is inherently non-convex and typically requires strong incoherence and high sensing rates for exact recovery. The authors propose converting the problem into a noisy matrix sensing formulation by injecting controlled noise perturbations, thereby ensuring valid Restricted Strong Smoothness (RSS) and Restricted Strong Convexity (RSC) conditions.

The paper first establishes conditions under which this surrogate problem, termed $\epsilon$-MC, admits a global solution close to the ground-truth matrix (Theorem 2.2). A key contribution is the derivation of an explicit accuracy–sampling probability trade-off, parameterized by the sampling rate $p \in [0,1]$ and the perturbation level $\epsilon$.

While this formulation guarantees valid RSS and RSC constants, it still suffers from large RIP constants, making optimization challenging.
To address this, the authors adopt and extend the lifted tensor framework of Ma et al. (2024) to the noisy setting, leveraging overparameterization to guarantee favorable optimization geometry—specifically, that all non-global critical points are strict saddles. Numerical experiments comparing the proposed tensor PCA solver with several baselines (e.g., BM factorization, SDP relaxation, spectral reweighting) demonstrate improved recovery success rates across both independent and structured sampling regimes.

**Strengths:**

* The paper introduces a perturbed matrix completion formulation that ensures valid RSS and RSC constants and provides an explicit quantitative relationship between recovery accuracy, sampling rate, and noise level.
* It extends the lifted tensor framework to the perturbed setting, proving guarantees for local minima and establishing that spurious solutions become strict saddles under the proposed formulation.
* The inclusion of numerical experiments—though the paper is largely theoretical—adds credibility and demonstrates the practical advantage of the proposed approach over multiple baselines.

**Weaknesses:**

* Scalability: The lifted-tensor solver is memory-intensive and currently applicable only to small-scale matrices. The paper does not discuss possible algorithmic strategies, approximations, or structural assumptions that could make the proposed method practical for large-scale problems.
* Information-theoretic limits: While the paper derives explicit performance–sampling trade-offs, it does not discuss how close these results are to information-theoretic recovery limits for matrix completion. A comparison or partial converse analysis could provide valuable insight into the tightness and fundamental optimality of the proposed bounds.

**Questions:**

* Are there potential directions that could make the proposed approach more scalable?
* Can the authors provide any converse-style analysis or information-theoretic lower bounds on recovery accuracy in terms of the sampling rate $p$?

---

> ### Author Response · Authors · 2025-11-19
> **Response to Official Review of Reviewer oKEv -- Part 1**
>
> Dear Reviewer oKEv,
>
> We sincerely appreciate the time and effort you have devoted to reviewing our work.
>
> ## Regarding Scalability
>
> Thank you for raising the question about scalability. We agree this is an important point, and we would like to explain how it could be potentially improved in two ways:
>
>
> ### 1. Tensor lifting as a guiding oracle for deterministic escapes (work in progress)
>
> In the current manuscript, the lifted objective reveals strict saddles and new descent directions that *do not exist* in the original matrix parameterization. These directions live in the higher-dimensional tensor space and, in general, do not admit a simple back-projection to a single matrix iterate.
>
> At the same time, we have been investigating conditions under which these high-dimensional escapes *do* admit meaningful lower-dimensional counterparts. The key idea is to use the tensor point *after one escape step* as a reference rather than attempting to follow the entire lifted trajectory. Concretely, for a spurious local minimum $\hat X$ satisfying (16) (or (13) for the general case), its lifting $\hat{\mathbf{w}} = \hat X^{\otimes \ell}$ becomes a strict saddle. If we take an escape step along the negative-curvature direction in the lifted space,
>
> $$
> \mathbf{w}\_{\text{escape}} = \hat{\mathbf{w}} + \eta\ \operatorname{vec}(u\_n q\_r^\top)^{\otimes \ell},
> $$
> the loss in tensor space strictly decreases for a suitable step size $\eta$.
>
> The central question is then: under what conditions does $\mathbf{w}_{\text{escape}}$ admit a “good” projection back to a matrix $\check X$ that *also* has lower objective value than $\hat X$? Our preliminary results suggest that $\mathbf{w}\_{\text{escape}}$ can be approximated (in spectral norm) by a rank-$r$ matrix
>
> $$
> \check{X} = \sum_{\phi=1}^{r-1} \sigma_\phi v_\phi q_\phi^\top + \sigma_r(\alpha v_r + \beta u_n) q_r^\top,
> $$
> with tunable $\alpha, \beta$ depending on $\eta$, such that
>
> $$
> || \operatorname{vec}(\check{X})^{\otimes \ell} - \mathbf{w}_{\text{escape}} ||_S \to 0 \quad \text{as} \quad \alpha^\ell \to 1,\ \sigma_r^\ell \beta^\ell \to \eta.
> $$
> We further derive a sufficient condition—expressed via an “Escape Feasibility Score” (EFS)—under which $f(\check X) < f(\hat X)$, guaranteeing a *deterministic escape* in the original matrix space once certain curvature and RIP-type conditions hold.
>
> Conceptually, it shows that there is a principled way to **design deterministic escape steps without actually using tensors**. Because the aforementioned framework is substantial and serves wide independent interests, we will publish it as a separate follow-up work. In the expected future publication, we will also discuss its implications on the $\epsilon$-MC framework proposed in this work. We believe the novelty of this manuscript mostly lies in liking MC to noisy MS ($\epsilon$-MC), and using over-parametrization to solve the noisy MS problem.
>
>
> ### 2. Efficient tensor optimization via low-rank structure
>
> We would also like to emphasize that the deterministic escape framework described above is **not the only** way to make lifted approach scalable. Even if one works directly in the tensor space, our framework is inherently compatible with efficient implementations because the tensors we optimize are generally extremely low-rank after a few iterations[1]. This makes it natural to use compressed tensor formats rather than materializing the full $n^\ell$ tensor.
>
> Two concrete directions are:
>
> 1. **Low-rank tensor manifolds (Tucker / TT) with Riemannian optimization.**
>    Tucker and tensor-train (TT) formats represent a high-order tensor using a small number of factor matrices/cores, reducing storage from $O(n^\ell)$ to approximately $O(\ell n r)$ or $O(\ell n r^2)$ when the tensor rank $r$ is small. Riemannian optimization methods on these manifolds (e.g., for low-rank tensor completion) update only the compressed factors, not the full tensor size [2,3]. In our setting, the lifted iterate can be stored and updated purely through such low-rank factors, while using gradients derived from our lifted objective.
>
> 2. **Randomized low-rank compression.**
>    Randomized algorithms for Tucker/TT decompositions can further reduce cost by computing approximate low-rank representations from sketches of the tensor rather than its full entries [4].
>
> In practice, leveraging these ideas would require a nontrivial amount of **software engineering**: careful design of data structures for factorized tensors, GPU-friendly kernels, and adaptive rank/tolerance control. As a single, theory-focused author, I have not yet implemented such an optimized engine; instead, the current experiments are run at moderate scale with optimized JAX code. However, from a methodological standpoint there is no inherent obstacle, and can be implemented with a small group of software engineers.

---

> ### Author Response · Authors · 2025-11-19
> **Response to Official Review of Reviewer oKEv -- Part 2**
>
> (Continued)
>
> ## Comparison with Information-Theoretic Limits
>
> We're glad that the reviewer asked about the comparison between our framework (Theorem 2.2) and existing information theoretic guarantees. A fully direct comparison is difficult because Theorem 3 of [5] and our Theorem 2.2 address different notions of “success”:
>
> - Theorem 3 of [5] gives a **minimum number of observations per column**, hence leading to a minimum observation ratio $p_{\mathrm{IT}}(\delta)$, that guarantees **exact** rank-$r$ completion (algorithm and complexity independent) with probability at least $1-\delta$.
> - Our theorem gives a **lower bound on the probability** that $||M^\dagger - M^*||\_F^2 \leq R$, and from this we can infer the minimal sampling probability $p_{\min}(\epsilon_{\text{noise}},R,\delta)$ needed to achieve that accuracy with probability at least $1-\delta$ (also without considering algorithm, but later we use the lifted framework (13) as explained in this manuscipt).
>
> To make this more concrete, we updated the code underlying Fig. 1 and summarized the resulting tradeoffs as the two tables below, for $\epsilon_{\text{noise}} = 10^{-4}$ and $10^{-3}$. Smaller $\epsilon$ is better for recovery guarantees, but poses greater challenge for computation. However, we found in practice that when $\epsilon \in [10^{-3}, 10^{-5}]$, the lifted framework works optimally. The number in the table represents the minimum sampling probability ($p_{\min}$ and $p_{\mathrm{IT}}$) for recovery. This comparison assumes $n=100$, since a smaller $n$ would make Thoerem 3 of [5] vacuous.
>
> **Comparison at $\epsilon_{\text{noise}} = 10^{-4}$.**
>
> |   Failure Prob ($\delta$) |   p_IT_Thm3 |   R=0.0001 |   R=0.0005 |   R=0.0010 |   R=0.0100 |   R=0.0500 |
> |--------:|------------:|-----------:|-----------:|-----------:|-----------:|-----------:|
> |  0.3000 |      0.8171 |     0.2162 |     0.0628 |     0.0500 |     0.0500 |     0.0500 |
> |  0.2000 |      0.8658 |     0.2183 |     0.0670 |     0.0500 |     0.0500 |     0.0500 |
> |  0.1000 |      0.9489 |     0.2226 |     0.0692 |     0.0500 |     0.0500 |     0.0500 |
> |  0.0500 |      1.0321 |     0.2247 |     0.0713 |     0.0500 |     0.0500 |     0.0500 |
> |  0.0100 |      1.2252 |     0.2311 |     0.0777 |     0.0543 |     0.0500 |     0.0500 |
>
> - Even for a tight recovery radius of $0.0001$, we only needed a $p$ at around $0.2$, whereas traditional theories needed at least $80 \\%$ observed even at a $30 \\%$ failure rate.
>
> **Comparison at $\epsilon_{\text{noise}} = 10^{-3}$.**
>
> |   Failure Prob ($\delta$) |   p_IT_Thm3 |   R=0.0001 |   R=0.0005 |   R=0.0010 |   R=0.0100 |   R=0.0500 |
> |--------:|------------:|-----------:|-----------:|-----------:|-----------:|-----------:|
> |  0.3000 |      0.8171 |        nan |     0.8489 |     0.7296 |     0.2162 |     0.0649 |
> |  0.2000 |      0.8658 |        nan |     0.8510 |     0.7338 |     0.2183 |     0.0670 |
> |  0.1000 |      0.9489 |        nan |     0.8553 |     0.7360 |     0.2226 |     0.0692 |
> |  0.0500 |      1.0321 |        nan |     0.8574 |     0.7402 |     0.2247 |     0.0734 |
> |  0.0100 |      1.2252 |        nan |     0.8638 |     0.7445 |     0.2311 |     0.0777 |
>
> - For our approach, a very small radii (e.g. $R=10^{-4}$) become essentially unattainable (NaN or very close to $1$), which is consistent with the increased noise level.
> - For moderate radii ($R = 10^{-2}$ or $5\cdot 10^{-2}$), $p_{\min}$ is again much smaller than $p_{\mathrm{IT}}(\delta)$ (e.g., around $0.21$–$0.23$ and $0.06$–$0.08$, respectively).
>
> Overall, these tables illustrate that **by relaxing the requirement of exact recovery to small reconstruction error**, our $\epsilon$-MC framework can simultaneously achieve low error and high recovery probability with significantly smaller sampling ratios. This is exactly the regime we focus on in our main experiments, and we hope this comparison clarifies how our guarantees relate to classical information-theoretic limits.
>
> **References (for both rebuttal comments)**
>
> [1] Z. Ma, J. Lavaei, and S. Sojoudi, “Algorithmic regularization in tensor optimization: Towards a lifted approach in matrix sensing,” *Advances in Neural Information Processing Systems*, vol. 36, 2024.
>
> [2] D. Kressner, M. Steinlechner, and B. Vandereycken, “Low-rank tensor completion by Riemannian optimization,” *BIT Numerical Mathematics*, vol. 54, no. 2, pp. 447–468, 2014.
>
> [3] S. Holtz, T. Rohwedder, and R. Schneider, “On manifolds of tensors of fixed TT-rank,” *Numerische Mathematik*, vol. 120, no. 4, pp. 701–731, 2012.
>
> [4] R. Minster, A. K. Saibaba, and M. E. Kilmer, “Randomized algorithms for low-rank tensor decompositions in the Tucker format,” *SIAM journal on mathematics of data science*, vol. 2.1, pp. 189-215, 2020.
>
> [5] Pimentel-Alarcón, D. L., Boston, N., and Nowak, R. D. (2016). A characterization of deterministic sampling patterns for low-rank matrix completion. *IEEE Journal of Selected Topics in Signal Processing*, 10(4):623-636.

---

### Official Review · Reviewer_svJS · 2025-10-30

**Soundness:** 3
**Presentation:** 4
**Contribution:** 3
**Rating:** 6
**Confidence:** 4

**Summary:**

The authors address the matrix completion problem and propose to reformulate it as a noisy matrix sensing problem. They establish theoretical conditions showing that the solution to this reformulated problem remains close to that of the original matrix completion formulation, and they further propose handling the problem through a lifted tensor framework.

**Strengths:**

Compared with existing approaches, the proposed method enables successful completion even when the sampling rate p is very small, though this comes at the cost of reduced accuracy. The idea is interesting, and the paper is well written, well structured, and clearly presented.

**Weaknesses:**

I have a concern regarding the experimental validation. The authors evaluate the success rate over 20 trials using a fixed estimation error threshold and a fixed noise level. Although an explanation is provided for this choice, the experimental setup, in my view, does not convincingly support the paper’s main claim—that the proposed method enables completion at low p values but sacrifices accuracy, especially with a user-chosen noise level. To substantiate this point, the experiments should be improved, either by including additional results (e.g., varying the noise level or error threshold) or by providing a more detailed justification and interpretation of the current setup.

**Questions:**

The analysis focuses solely on the overall estimation error. It would be interesting to explore where the reconstruction errors occur relative to the ground truth. Are these errors concentrated in specific regions of the estimated matrix, or are they more uniformly distributed?

---

> ### Author Response · Authors · 2025-11-19
> **Response to Official Review of Reviewer svJS -- Part 1**
>
> Dear Reviewer svJS,
>
> We sincerely appreciate the time and effort you have devoted to reviewing our work, and we value the opportunity to further clarify our experimental findings.
>
>
> ## **Further Clarifications on Experiments**
>
> We first clarify our choice of using *success rate* rather than normalized reconstruction error. Consider the normalized reconstruction errors from ten trials in the Fig. 2(a) experiment:
>
> | Trials | 1 | 2 | 3 | 4 | 5 | 6 | 7 | 8 | 9 | 10 |
> |-------|---|---|---|---|---|---|---|---|---|----|
> | BM Approach | 2.82 | 2.00 | 0.01 | 2.00 | 2.00 | 2.00 | 2.00 | 2.00 | 2.00 | 2.00 |
> | Our Approach (lifted) | 8.37e-5 | 5.81e-4 | 2.26e-4 | 8.19e-5 | 8.3e-6 | 4.00 | 3.97e-4 | 8.21e-5 | 8.3e-6 | 3.5e-5 |
>
> As shown, our lifted approach solves nearly all instances extremely accurately, except for rare outliers (e.g., trial 6). If we use mean construction error, it will equal to *0.4* here with standard deviation of 1.2, which fails to capture the true moral of story given high discrepancies between successful and failed trials. As the reviwer also suggested, we have added **Appendix E** to further analyze why reconstructions errors look like this.
>
> Following the reviewer’s suggestion for additional empirical evidence, we added a few comparative studies on both the user-chosen noise parameter $\epsilon$ and the sampling probability $p$. These results now appear in a new **Appendix B** (with the original Appendix B becoming Appendix C). Please refer to the new appendix for new plots and below we present the results as tables.
>
>
> ## **Effects of $\epsilon$**
>
> Under the setting of (20) with $n=6$, initialization magnitude $10^{-4}$, $\ell = 3$ (matching Fig. 2(a)), we ran **100 trials** (instead of 20) for each $\epsilon$:
>
> | $\epsilon$ value | BM Success (ref)| Success ($\le 0.05$) | Success ($\le 0.01$) |
> |-----------------:|----------------------:|-----------------------:|-----------------------:|
> | 0.1              | 0.09| 0.00                  | 0.00                  |
> | 1e-2             | 0.09| 0.02                  | 0.02                  |
> | 1e-3             | 0.09| 0.08                  | 0.08                  |
> | 5e-4             |0.09|  0.14                  | 0.14                  |
> | 1e-4             | 0.09| 0.23                  | 0.21                  |
> | 5e-5             |0.09|  0.36                  | 0.35                  |
> | 1e-5             | 0.09| 0.11                  | 0.11                  |
>
> Interpreting these results through Theorem 4.1 and (16): the error bound has a *computational part* which is $\frac{1}{\epsilon}\lambda_r(\hat X)\sqrt{\mathrm{tr}(M^\star)}$, and an *accuracy part* $e_1$ characterizing the perturbation between $\epsilon$-MC and the original MC problem. As explained further in our response to Reviewer py6p and Lemma D.1, the constant $\frac{1}{\epsilon}$ is conservative. Nevertheless, decreasing $\epsilon$ simultaneously improves accuracy and worsens conditioning. When $\epsilon$ is large (0.1, 0.01), $e_1$ dominates, leading to poor performance even under global convergence. When $\epsilon$ decreases to moderate levels (1e-3 to 5e-5), both terms balance, producing the best performance. When $\epsilon$ becomes too small (1e-5), the computational difficulty dominates and success rate deteriorates. As described in Appendix E, successful recoveries normally have very small reconstruction errors, so the choice of success threshold only minimally affects overall success rate, as evident in our table.
>
>
> ## **Effects of sampling probability $p$**
>
> Adopting the setting in Fig. 2(b) ($n=6$, $\epsilon$=5e-4 init = $10^{-4}$, $\ell=3$), we tested MC problems under independent random observation of each entry. For each level of $p$, we randomly selected 10 masks, each performing 20 trials in total (200 trials in total) for both the unlifted BM approach and lifted $\epsilon$-MC approach:
>
> | $p$    | BM success | Lifted $\epsilon$-MC success |
> |--------|------------|------------------------------|
> | 0.05   | 0.00       | 0.01                         |
> | 0.10   | 0.1       | 0.2                          |
> | 0.15   | 0.07       | 0.24                         |
> | 0.20   | 0.22      | 0.28                         |
> | 0.25   | 0.31       | 0.44                         |
> | 0.30   | 0.4       | 0.58                         |
> | 0.40   | 0.45       | 0.6                         |
>
> As $p$ increases, success generally improves for both methods. Our method provides the clearest advantage in low-$p$ regimes where BM struggles, while at high $p$ the baseline already achieves many successes, thus narrowing the margin. If we again explain this phenonmenon using (16), we find that $e_1$ can be effetively reduced with higher $p$, whereas the reduction of the computational part is not directly tied to $p$.

---

> ### Author Response · Authors · 2025-11-19
> **Response to Official Review of Reviewer svJS -- Part 2**
>
> (Continued from Part 1)
>
>
> ## **Regarding Concentration of Errors**
>
> We may not have fully captured the reviewer’s intended meaning, but our interpretation is that the question concerns *where* the reconstruction errors appear within the matrix—i.e., the spatial distribution of  $E = M^{\text{sol}} - M^\star$—rather than only the magnitude $||E||_F$.
>
> Our theoretical framework focuses on the landscape of the lifted tensor objective and characterizes when gradient-based methods escape spurious local minima. Although it ensures small global $||M^{\text{sol}} - M^\star||_F$, it does not describe the entrywise distribution of $E\_{ij}$.
>
> To address this, the newly added **Appendix E** now includes a detailed numerical study. For the structured instance (20), we examine the entrywise squared error
>
> $$
> E_{ij}^2 = (M^{\text{sol}}\_{ij} - M^\star\_{ij})^2
> $$
>
> Our main findings:
>
> - The error on **observed** entries is essentially zero in all runs.
> - In unsuccessful runs, all noticeable error lies on **unobserved** entries. The optimization interpolates observed data perfectly but converges to distinct benign local minima on the missing part.
> - The regions of error on unobserved entries vary significantly across runs, with no consistent row, column, or block repeatedly failing. This matches the theoretical prediction that (20) contains a large set of spurious minima.
> - Successful runs show uniformly small error everywhere.
>
> For a more detailed view of the spatial patterns of the errors (including how they relate to the sampling mask), we refer the reviewer to the four heatmaps added in Appendix E of the revised manuscript.
>
> ## **Final Remarks**
>
> Once again, we thank you for your thoughtful and constructive review. We hope this rebuttal addresses your concerns and clarifies some confusions. If this response resolves some of your reservations, we would sincerely appreciate your reconsideration of our rating.

---

> > ### Comment · Reviewer_svJS · 2025-11-25
> >
> > Thank you for the response. The additional experiments and analysis significantly strengthen the paper. I will adjust my score accordingly.

---

> > > ### Author Response · Authors · 2025-11-26
> > >
> > > Dear Reviewer svJS:
> > >
> > > Thank you for the time and effort dedicated towards reviewing our work, and for recognizing its value. Your comments and suggestions have indeed helped us to improve the quality of this work.

---

### Official Review · Reviewer_py6p · 2025-10-31

**Soundness:** 3
**Presentation:** 3
**Contribution:** 2
**Rating:** 4
**Confidence:** 2

**Summary:**

The author studies how to inject noise perturbation to turn matrix completion problem into a noisy matrix sensing problem, which can be easier to solve. They show that the induced matrix sensing problem has a  RIP property and establish the accuracy-probability trade-offs for the noisy matrix sensing problem. Then, they employ the lifted tensor framework to further deal with the matrix sensing problem, to overcome the small RSC constant issue.

**Strengths:**

1. The paper is well-structured and easy to follow. The high-level motivation and the intuitive explanation about the main theoretical results are provided in a clear way.

2. The theoretical contributions are solid. The idea of introducing perturbation noise into a matrix completion problem and reformulating it as a noisy matrix sensing problem is novel. The paper also presents interesting theoretical results, such as the probability–accuracy trade-off, which is a new contribution in this area based on my knowledge.

**Weaknesses:**

The main concern lies in the interpretation of the derived theoretical results. In Theorem 3.1, the upper bound includes the term $1/\alpha_s$. However, since the transformed matrix sensing problem only has a small RSC constant, $\alpha_s=\varepsilon^2$ could be very small. In fact, in your experiment, the $\varepsilon$ is set to be $1e-5$, which would render the theoretical bound practically meaningless. Similar concerns apply to Theorem 4.1, where the upper bound also contains a $1/\varepsilon$ term, which is also very large when $\varepsilon$ is set to be 1e-5.

Then, if the perturbed noise matrix sensing problem has a small RSC constant and will finally lead to a practically meaningless result, the advantage of using this perturbation becomes unclear.

**Questions:**

1. Could the authors clarify that why  the proposed approach still outperforms other baselines even if $\varepsilon$ is set to be 1e-5 in the experiment? Does this suggest that the theoretical results may leave substantial room for improvement?

---

> ### Author Response · Authors · 2025-11-19
> **Response to Official Review of Reviewer py6p**
>
> Dear Reviewer py6p,
>
> We sincerely appreciate the time and effort you have devoted to reviewing our work, and we value the opportunity you have given us to elaborate on one subtlety that we glossed over in our theoretical derivations.
>
> ---
>
> **Regarding Tightness of Lemma 2.1**
>
> We first acknowledge that the reviewer was correct in pointing out that the lower bound in Lemma 2.1 is quite conservative. Indeed, our proof of Lemma 2.1 relies only on the simple inequality
> $$
> ||\mathcal{A}\_{\Omega,\epsilon}(M)||\_2^2
> = \epsilon^2 ||M||\_F^2 + (1-\epsilon^2)||M||\_{\Omega,F}^2
> \ge \epsilon^2 ||M||\_F^2,
> $$
> which intentionally neglects the random variable $(1-\epsilon^2)||M||\_{\Omega,F}^2$. While this bound is very conservative, incorporating the $||M||\_{\Omega,F}^2$ term would require assumptions on $M$, which would undermine the RIP property’s global applicability. For instance, as illustrated in the toy example at the beginning of Section 2, there are many cases in which $||M||\_{\Omega,F}^2 = 0$, making the lower bound $\epsilon^2||M||\_F^2$ tight when considering all possible $M$. Such generality is crucial because the RIP property is applied in the proof of Theorem 3.1 to the matrix $\hat X \hat X^\top - M^*$, whose structure cannot be predetermined, since $\hat X$ may be any critical point.
>
> That said, this conservative bound does not significantly affect the main result (16). In (16), we simplified $||\tilde w||\_2^2 / \alpha\_s$ in (13) to $e\_1$ by substituting expressions for both $||\tilde w||$ and $\alpha\_s$, thereby eliminating the $1/\epsilon^2$ term and obtaining
> $$
> e\_1 = \sqrt{ \frac{1-p+\chi}{p-\chi}} \epsilon ||M^*||\_F.
> $$
>
> As a result, choosing a small $\epsilon$ does not degrade the bound on $||M^* - \hat X \hat X^\top||\_F$ in an inversely quadratic manner. Moreover, in practice the true value of $||\mathcal{A}\_{\Omega,\epsilon}(M)||\_2^2$ is much larger than $\epsilon^2||M||\_F^2$, a fact confirmed (and pointed out by the reviewer) by observing that even a very small choice such as $\epsilon = 5 \times 10^{-5}$ produced favourable results. Thus, although we required a clean lower bound for theoretical completeness, the experiments validate the use of small $\epsilon$ constants.
>
> Nevertheless, we agree with the reviewer that Lemma 2.1, when invoked in isolation, appears too weak in its current form. Below, we outline how the bound can be strengthened by assuming i.i.d. observation of entries and by further relating the RIP constant to the spikiness ratio of the matrix $M$.
>
> ---
>
> **Improving Lemma 2.1**
>
> Here we outline how a tighter bound that takes into consideration the structure of $M$ can be obtained, and we have **revised our manuscript** to include a **formal version of this result as Lemma D.1 in the appendix** for the reviewer's perusal.
>
> First we clarify our setting:
> - Each entry $(i,j)$ is observed independently with probability $p \in (0,1]$.
>   - Let $S\_{ij} \sim \mathrm{Bernoulli}(p)$ i.i.d.
>   - Define the sampling operator
>     $$
>       (P\_\Omega(M))\_{ij} := S\_{ij} M\_{ij}.
>     $$
> - The number of observations is $m = \sum\_{i,j} S\_{ij}$, so $\mathbb{E}[m] = p n^2$.
>
> We use the entrywise max norm
> $$
> ||M||\_{\max} := \max\_{i,j} |M\_{ij}|.
> $$
>
> Define the **spikiness** of $M$ as
> $$
> \alpha\_{\mathrm{sp}}(M)
> := \frac{\sqrt{N}\,||M||\_{\max}}{||M||\_F}
> = \frac{n\,||M||\_{\max}}{||M||\_F}, \quad N = n^2.
> $$
>
> Intuition:
>
> - If entries are all similar in size, then $||M||\_{\max} \approx ||M||\_F / n$ and $\alpha\_{\mathrm{sp}}(M) \approx 1$.
> - If $M$ has a few very large entries, then $||M||\_{\max} \gg ||M||\_F / n$ and $\alpha\_{\mathrm{sp}}(M)$ is large.
> - Deterministically, $1 \le \alpha\_{\mathrm{sp}}(M) \le n$.
>
> Then from there we could apply scalar-wise Berstein to each $S_{ij}$ to control deviation from mean and obtain (please see Appendix D for full proof), with probability at least $1-\delta$:
>
> $$
> \frac{||P\_\Omega(M)||\_F^2}{||M||\_F^2} \ge p - \alpha\_{\mathrm{sp}}(M)^2 \frac{(1+2p)\,\log(1/\delta)}{n^2}
> $$
>
> Finally, combining everything, we have with probability at least $1-\delta$,
>
> $$
> ||\mathcal{A}\_{\Omega,\epsilon}(M)||\_2^2 \ge  \Big( \epsilon^2 + (1-\epsilon^2) \Big[p-\alpha\_{\mathrm{sp}}(M)^2 \frac{(1+2p)\,\log(1/\delta)}{n^2}\Big] \Big)\, ||M||\_F^2.
> $$
>
> Therefore, in expectation the effective RSC constant is
>
> $$
> p + \epsilon^2 (1-p),
> $$
> as opposed to $\epsilon^2$ in the worst case and the above concentration inequality controls the deviation from this mean using the probability $\delta$. For non-spiky matrices $M$ with $\alpha\_{\mathrm{sp}}(M) \ll n$, the deviation from its mean $p + \epsilon^2 (1-p)$ will be relatively small and can often be ignored.
>
> ---
>
> **Final Remarks**
>
> Once again, we thank you for your thoughtful and constructive review. We hope this rebuttal addresses your concerns and clarifies some confusions. If this response resolves some of your reservations, we would sincerely appreciate your reconsideration of our rating.

---

### Author Response · Authors · 2025-12-03
**A Message to Area Chairs**

Dear Area Chairs:

We again extend our gratitude to the Area Chairs and Reviewers who devoted their valuable time to reviewing our work. In light of the chaotic events that happened recently pertaining to identity leakages, we hope to provide a succinct summary of the rebuttal process in order to facilitate more efficient evaluation of this work by new ACs, who may not have had the chance to go over our work in details given the demanding workload.

In the rebuttal process, we have uploaded a new version of this work, and have addressed the following problems:

1. We have added Lemma D.1 to strengthen Lemma 2.1, which directly addresses Reviewer py6p's concern. We have also explained in our rebuttal why Lemma 2.1, as it appears in the main text, is actually tight over all matrices, but slack on average. Unfortunately reviewer py6p did not have a chance to respond to our rebuttal.

2. We have added further ablation studies for our experiments in the new Appendix B, and have analyzed the solution structures in the new Appendix E. This directly addresses Reviewer svJs's concerns and he adjusted his score to "Accept" following our response.

Moreover, we have explained to Reviewer oKEv in detail how we could improve the scalability of this approach, and have performed new experiments to show that this approach requires a much lower sampling probability $p$ than existing information theoretic limits under similar settings, which demonstrates the power of this framework by allowing slight corruptions in solution accuracy. However, Reviewer oKEv also did not get a chance to respond to our response, which is unfortunate.

Overall, we believe our response has addressed the major concerns of the reviewers, and the only reviewer who responded (svJs) also increased his rating as the result. Our goal is to give the ACs an executive summary of this process, and leave our work to the ACs best judgements.

---

### Meta-Review · Area_Chair_rTVs · 2026-01-06

**Summary:**

Reviewers agree the paper is technically careful and proposes an interesting reframing of scarce-observation matrix completion via a noise-perturbed surrogate plus a lifted-tensor optimization argument.  However, the following weaknesses were highlighted:

* Unclear motivation and practical interpretability of the theory,
* Limited scalability of the proposed practical approach (high-order tensor lifting is extremely memory-intensive),
* Remaining novelty and positioning questions, i.e. what is substantively new versus an integration of existing ideas, and whether the contribution is significant relative to a large existing literature.

In my own reading, I share the last concern and think it is central. Since the paper is explicitly attacking a well-established, heavily studied problem, the bar for "contribution" is naturally high. It is not enough to be different, but it should be clear in what sense it is objectively better (setting, guarantees, rates, constants, regimes, or empirical wins that persist under fair baselines). The paper's headline positioning is that, unlike prior work, it can work with an "arbitrary $p$" via an accuracy-probability trade-off. The paper gives the disclaimer that it cannot achieve "arbitrary accuracy" with arbitrary $p$, but even then the claim reads too strong and risks being misleading. Below info-theoretic identifiability thresholds, completion is not merely "less accurate," it can be arbitrarily wrong because the unseen entries can go to infinity while remaining consistent with the observations. Any finite error guarantee must either become vacuous or must impose additional restrictions (bounded entry magnitudes, distributional assumptions, spikiness/incoherence-type control, etc.).

Relatedly, even accepting the surrogate framing ("recover $M^\dagger$ close to $M^*$" rather than exact recovery), what is the remaining error rate once the analysis is carefully disentangled into what is possible versus impossible? Concretely, after specializing assumptions so that the bound is non-vacuous, how does the resulting error scale with $p,n,r,\mu,\kappa$ (and with the noise/perturbation parameter), and is that scaling meaningfully better than existing theory for noisy completion/sensing surrogates or regularized methods?

I think the paper could merit publication if the authors significantly strengthen the contribution narrative and positioning. It should clearly state the regime of validity, explicitly acknowledge where bounds must become vacuous, and provide an apples-to-apples comparison (theoretical and/or empirical) that demonstrates a concrete advantage over established baselines in a well-defined setting.

**Reviewer Concerns:**

Theory usefulness and tightness. Bounds depend on small RSC/RIP-type constants. Guarantees are effectively vacuous at the experimental settings.

Scalability and practicality. Lifted-tensor solver appears memory intensive and only demonstrated for small matrices. Unclear how to make useful for practice.

Novelty/Positioning. Need clearer comparison to matrix completion information-theoretic limits. What does successful recovery (approx vs exact) mean?

**Reviewer Scores:**

Perhaps positive scores will become more positive, but the negative score is likely unchanged.

---

> ### Public Comment · ~Ziye_Ma1 · 2026-04-02
> **Respectful Disagreement with the AC's comment**
>
> First and foremost, I want to extend my gratitude towards the AC and reviewers for spending their precious time to read, review and help revise my work as author. Especially, I totally acknowledge the heavy burdens that are shouldered by the ACs, given this year's special circumstances, and is highly appreciative of their voluntary contribution to the ML community.
>
> However, I respectfully and strongly disagree with the AC's meta review this year, given the careless reading of the paper and unprofessional attitude towards the peer review process.
>
> Firstly, I hope to point out that I also agree with the AC's saying that any recovery guarantee under information theoretic limits "is not merely less accurate, it can be arbitrarily wrong because the unseen entries can go to infinity". And this is exactly why Theorem 2.2 exists. Instead of using a DETERMINISTIC result, we presented PROBABILISTIC results to smooth out the unprobable event that one or two unseen entries will go to infinity (or become very very large) in real-life settings. Our results state that both the recovery accuracy and the probability under which such guarantee holds all depend on $p$, which is intuitive to understand. As the number of observed entries decrease ($p$ decreases), not only will the accuracy worsen, the probability that this bound holds will also drop dramatically, which is perfectly in-line with the AC's observation. Moreover, to make sure that our probabilistic bounds are not vacuous, we plotted figure 1 to show readers that even with very low $p$ (even $p$ below information theoretic limits), the bound is tight and holds with high probability. Therefore, our results are not misleading in the sense that we claimed a universal accuracy that always holds under information theoretic limits, which will be impossible.
>
> Secondly, the AC asked for "how does the resulting error scale with $p,n,r,\mu,\kappa$ ", and this is something we precisely included in (19) of Theorem 4.1, our main result. Given the application of the very powerful over-parametrization framework (Theorem 3.1), we are able to get rid of the dependency on $\mu$ and $\kappa$, which is the key innovation in this work. As for $p$, it is the main deciding factor, and (19) describes how this error will increase as $p$ decreases, which we just studied in Theorem 2.2 in detail. I think the AC might have just skimmed through our paper and had some confusions towards our claims and decided to directly jump to the final conclusion without carefully reading our results, since it would be hard to miss otherwise.
>
> Finally, and most importantly, the AC is showing great disrespect towards the peer review process itself, since the AC's job is not to carefully read the paper, but to moderate the reviewers' opinions and filter out long-tail responses. I appreciate the AC's effort to read our paper and give us his/her own honest opinion, but simply summarizing the months-long review process as "Perhaps positive scores will become more positive, but the negative score is likely unchanged." is highly unprofessional and arrogant. All three reviewers had carefully read my work, two of them liked it a lot, and one was not really sure (given his confidence score). Given the special circumstances, the one unsure reviewer didn't have time to respond, and saying something like "negative score is likely unchanged" is purely personal bias. In fact, most of the AC's meta review contents are not from the actual reviews if anyone would read it carefully, and most of it is the AC's own review. Sure, the AC has the final verdict on a paper, but if he/she simply decides to ignore all reviewers, why not just skip the peer-review process since it could all save us at least four month's worth of effort?
>
> Overall, I find this particular meta-review to be very disappointing and unprofessional. This does not mean that I believe the AC is not competent or has any ethical issues. Given the extreme conditions this year, I totally understand that mistakes/oversights will happen, and I just hope to point it out to give my work justice.

---

### Decision · Program_Chairs · 2026-01-26

Reject